

# Change in grounding line location on the Antarctic Peninsula measured using a tidal motion offset correlation method

Benjamin J. Wallis[1], Anna E. Hogg[1], Yikai Zhu[2,3], Andrew Hooper[2]

[1]School of Earth and Environment, University of Leeds, United Kingdom
[2]COMET, School of Earth and Environment, University of Leeds, United Kingdom
[3]Chinese Antarctic Centre of Surveying and Mapping, Wuhan University, China

*Correspondence to*: B.J. Wallis (eebjwa@leeds.ac.uk)

**Abstract**: The grounding line position of glaciers and ice shelves is an essential observation for the study of the Earth's ice sheets. However, in some locations, such as the Antarctic Peninsula, where many grounding lines have not been mapped since
the 1990s, remote sensing of grounding line position remains challenging. Here we present a tidal motion offset correlation (TMOC) method for measuring the grounding line position of tidewater glaciers and ice shelves, based on the correlation between tide amplitude and synthetic aperture radar offset tracking measurements. We apply this method to the Antarctic Peninsula Ice Sheet to automatically delineate a new grounding line position for 2019-2020, with near complete coverage along 9,300 km of coastline, updating the 20-year-old record. A comparison of the TMOC grounding line to contemporaneous
interferometrically-measured grounding line position shows the method has a mean seaward offset compared to interferometry of 185 m and a standard deviation of 295 m. Our results show that over the last 24 years there has been grounding line retreat at a number of fast flowing ice streams on the Antarctic Peninsula, with the most retreat concentrated in the north-eastern sector, where grounding lines have retreated following the collapse of ice shelves. We observe a maximum grounding line retreat since 1996 of 16.3 km on Hektoria Glacier, with other notable glaciers retreating by 9.3 km, 9.1 km, and 3.6 km
respectively. Our results document dynamic change on Antarctic Peninsula glaciers and show the importance of using an updated grounding line location to delineate the boundary between floating and grounded ice.

## 1 Introduction

The boundary between grounded ice resting on bedrock or sediments and floating ice, the grounding line (GL), is a key glaciological parameter essential for understanding the behaviour of marine terminating land ice. The GL location influences
ice dynamics because it marks the transition between an inland flow regime controlled by vertical shear stress and basal drag, and a frictionless floating ice flow regime controlled by longitudinal stresses. Accurate knowledge of the GL is a boundary condition required for both modelling of ice sheets and glaciers in small domain specialised models and large Earth system models used to project future ice sheet behaviour and sea-level rise contributions (Vieli and Payne, 2005; Pattyn et al., 2006; Pattyn, 2018; Cornford et al., 2020), and for calculations of observational datasets including ice shelf basal melting (Rignot et
al., 2013; Paolo et al., 2015; Gourmelen et al., 2017) and ice mass discharge, which is required as an input dataset for mass



balance calculations using the input output method (Mouginot et al., 2014; Gardner et al., 2018; Rignot et al., 2019; Davison et al., 2023). Change in the GL location over time is a sensitive indicator of ice sheet mass balance and stability, where GL retreat is associated with increased ice discharge and ice sheet mass loss (Joughin et al., 2012; Rignot et al., 2014; Joughin et al., 2016). Furthermore, GL retreat on retrograde bed slopes can cause a positive feedback leading to further retreat through
the process of marine ice sheet instability (Schoof, 2007).

Rather than having a fixed location, the grounding line is a transitory feature which constantly changes over short (daily) and longer term (decadal) timescales. It is located within a wider grounding zone which characterises the larger area (1 – 10 km wide) where the transition from grounded to complete hydrostatic equilibrium occurs (Smith, 1991; Vaughan, 1994; Fricker et al., 2009; Brunt et al., 2010, 2011). Within this zone the true GL where grounded ice loses contact with the bed can
migrate with changing sea-level caused by ocean tides and atmosphere pressure variations by the inverse barometer effect (IBE) and the extent of this migration is additionally controlled by bed topography, ice thickness and ice rheology (Jonathan and R, 1994; Brunt et al., 2010; Padman et al., 2018). This is further complicated by non-linear tidal migrations which can show threshold and hysteresis behaviour (Milillo et al., 2022; Freer et al., 2023). The grounding zone is made up of several features; the most inland of these is the landward limit of ocean induced ice flexure, point F, also known as the hinge line,
which is located slightly inland of the true GL, point G, due to the elastic properties of ice (Vaughan, 1994; Rignot et al., 2011a; Padman et al., 2018). In the seaward direction this is followed by the break in surface slope, point $I_b$, and the landward limit of stable hydrostatic equilibrium, point H. Additionally, in locations where there is an ice plain at the grounding zone, point $I_b$ may be located inland of the GL, point G (Corr et al., 2001; Brunt et al., 2011). For the purposes of this study, we use the term 'grounding line' to refer to the inland limit of the grounding zone identified by remote sensing methods, and we are
explicit about which grounding zone feature this refers to where required.

Several methods exist for measuring grounding line location and mapping grounding zone features from satellite remote sensing data. The most accurate method is to use synthetic aperture radar interferometry (InSAR) (Goldstein et al., 1993; Joughin et al., 2010a; Rignot et al., 2011a; Joughin et al., 2016; Rignot et al., 2016), where repeat-pass, phase sensitive synthetic aperture radar (SAR) measurements are used to create interferograms, which measure phase difference in the radar
line of sight. Interferograms of the grounding zone show fringing caused by horizontal ice flow displacements, with the GL visible as a dense pattern of fringes, which represent phase change caused by large vertical ocean tide. When two interferograms are differenced, fringes caused by steady state ice flow that are consistent in both measurements are removed to form a double-difference interferogram, with the remaining fringes caused by vertical displacement of the floating ice surface due to ocean tides, which do not remain constant over the same time period (Rignot, 1996). The inland limit of these
fringes denotes the inland limit of ice tidal flexure, point F.

While highly accurate, the major limitation of the differential SAR interferometry (DInSAR) technique for grounding line measurements is the requirement for coherence between SAR acquisitions. This particularly impacts measurements in locations where the ice surface changes rapidly, for example due to precipitation and surface melting, or in regions of fast ice flow and high deformation. Other remote sensing techniques that do not rely on InSAR coherence have also been used to map



GL positions; these include measuring tidal elevation change through repeat measurements by altimeter satellite instruments ICESat (Fricker and Padman, 2006; Fricker et al., 2009; Brunt et al., 2010; Xie et al., 2016), ICESat-2 (Li et al., 2020, 2022), and CryoSat-2 (Dawson and Bamber, 2017, 2020), which along with DInSAR GL fringe delineation are dynamic methods that locate point F and other grounding zone features by measuring vertical ice motion in response to short-term local sea-level variation. A small number of studies have used a technique called differential range offset tracking (DROT), a dynamic

technique which measures vertical tidal motion in SAR imagery through intensity feature tracking rather than interferometry, to observe point F on individual glaciers of interest in regions without interferometric coherence (Joughin et al., 2010b; Marsh et al., 2013; Hogg, 2015). The performance of the method is dependent on the radar frequency which determines the sensitivity to change in slant range path length caused by vertical displacement of the ground surface, and the magnitude of the tide amplitude in the study region. GL position measurements can also be made using static methods which do not measure ice

motion, for example by measuring the break in surface slope, point $I_b$, either from radar altimetry (Partington et al., 1987; Herzfeld et al., 1994; Jonathan and R, 1994; Fricker et al., 2000; Hogg et al., 2018), laser altimetry (Herzfeld et al., 2008; Bindschadler et al., 2011) and DEMs (Stearns, 2011; Rott et al., 2020), or from shading in optical satellite images (Scambos et al., 2007; Bindschadler et al., 2011).

In the satellite remote sensing era, these techniques, or combinations thereof, have been used to map grounding line

position and change across the Antarctic Ice Sheet (AIS). DInSAR mapping of the GL in Antarctica has been carried out using SAR data since 1992 (Goldstein et al., 1993), producing GL datasets for the entire continent from a combination of satellites (Rignot et al., 2011b, 2013, 2016; Mouginot et al. 2017). Additionally, in 2018, GL maps in Antarctica from Sentinel-1 SAR data were produced using deep learning to automate the fringe delineation process (Mohajerani et al., 2021), greatly increasing the volume of GL data produced by removing the need for manual delineation. Repeat track laser altimetry has been used to

map grounding zone features in Antarctica from ICESat-2 data (Li et al., 2020, 2022) complementing earlier studies, and radar altimetry from CryoSat-2 has produced continent-wide GL and grounding zone feature mapping through both a dynamic pseudo-radar-crossover approach (Dawson and Bamber, 2020) and a static surface slope approach (Hogg et al., 2018). Photoclinometry which measures the change in surface slope from surface shading has been used to measure the GL, and was combined with laser altimetry in the Antarctic Surface Accumulation and Ice Discharge (ASAID) project to produce continent

wide GL and hydrostatic line dataset (Bindschadler et al., 2011). There is a recognised requirement for monitoring the location of ice sheet GL's due to their importance as an indicator of ice sheet stability, for the interpretation of other observations, and modelling glacier and ice sheet behaviour (Joughin et al., 2012; Bojinski et al., 2014). However, despite significant progress, no method provides continuous monitoring of the GL around the whole Antarctic coastline with high-spatial sampling at regular time intervals.

Here, we develop a new method for measuring the grounding line location, which we call tide motion offset correlation (TMOC), this uses correlation of Sentinel-1 range direction intensity feature tracking data with an ocean tide mode, and apply it to measure the GL location on the Antarctic Peninsula (AP) in 2019-2020. The AP consists of an ice sheet-covered mountainous spine running north to south, with a coastline fringed by ice shelves totalling 110,000 km$^2$ in area in 2021 (Greene





et al., 2022). The ice shelves are clustered on the east and southwest coasts and, by contrast, the west coast north of 70°S is
dominated by tidewater glaciers. Observed ice mass losses on the Peninsula have been substantial; ice shelves lost an area of
28,000 km$^2$ between 1947 and 2008 (Cook and Vaughan, 2010), a trend which continued with observed losses of 20,500 km$^2$
between 1997 and 2021 (Greene et al., 2022). The collapse of ice shelves has increased ice discharge from glaciers that were
formally buttressed by the shelf (Rignot et al., 2004; Scambos et al., 2004; Rott et al., 2011; Seehaus et al., 2016; Friedl et al.,
2018) and the widespread retreat of tidewater glaciers has been linked to warming ocean temperatures on the west coast (Cook
et al., 2016). Overall, from 1992 to 2020 the Antarctic Peninsula Ice Sheet (APIS) has been responsible for 14 % of Antarctica's
total sea-level rise contribution (Shepherd et al., 2018; Otosaka et al., 2023).

Despite the AP being a highly dynamic and rapidly changing region of Antarctica, grounding line location
measurements are sparce in time, and few direct observations of GL change have been made because the region poses
significant challenges for established GL measurement methods. For DInSAR there are a number of areas of the AP which
have persistently low InSAR coherence. For repeat altimetry-based methods the ground track spacing is largest at lower latitude
locations such the AP which reduces the spatial resolution of GL products, and persistent cloudy weather limits the number of
successful laser altimetry retrievals. For these reasons, many parts of the AP coastline have not been measured using high-
precision tide-sensitive GL methods since the Ice and Tandem Phases of ERS-1/2 in 1991-1996, or from a combination of
static photoclinometry (1999-2003) and repeat track laser altimetry (2003-2008) from the ASAID GL product. More recent
GL measurements have been made for case studies of individual glaciers (Friedl et al., 2018; Rott et al., 2020), however, these
examples are rare, and the majority of the over 860 glaciers on the AP remain un-surveyed for over 20 years.

## 2 Tidal Motion Offset Correlation (TMOC) Method

### 2.1 Physical Basis

The existing DROT method exploits the off-nadir viewing geometry of SAR sensors to measure motion of the ice surface in
the range direction that is induced by the vertical displacement of floating ice with different ocean tide amplitudes between
image acquisitions. DROT utilises intensity feature tracking ice speed measurements in the range direction. A minimum of
two range direction speed maps are produced using 3 or 4 separate SAR acquisitions, and the speed maps are subsequently
differenced to remove the constant horizontal ice flow component, which leaves only the motion due to vertical movement of
the ice between acquisitions plus measurement noise. DROT has several limitations; it is significantly less sensitive to vertical
motion than DInSAR and hence places point F seaward of the DInSAR measured position, because the tidal displacement
signal must be greater than the measurement noise of feature tracking; and like DInSAR, individual results may not sample
the full range of tidal motion, and results must still be manually interpreted and delineated (Friedl et al., 2020).

We present a newly developed technique, Tidal Motion Offset Correlation (TMOC), for measuring the grounding
zone of glaciers and ice shelves using the same measurement principle as DROT, but substantially extending the methodology
with a time-series approach. The method measures the correlation between the range direction speed time-series and a time-





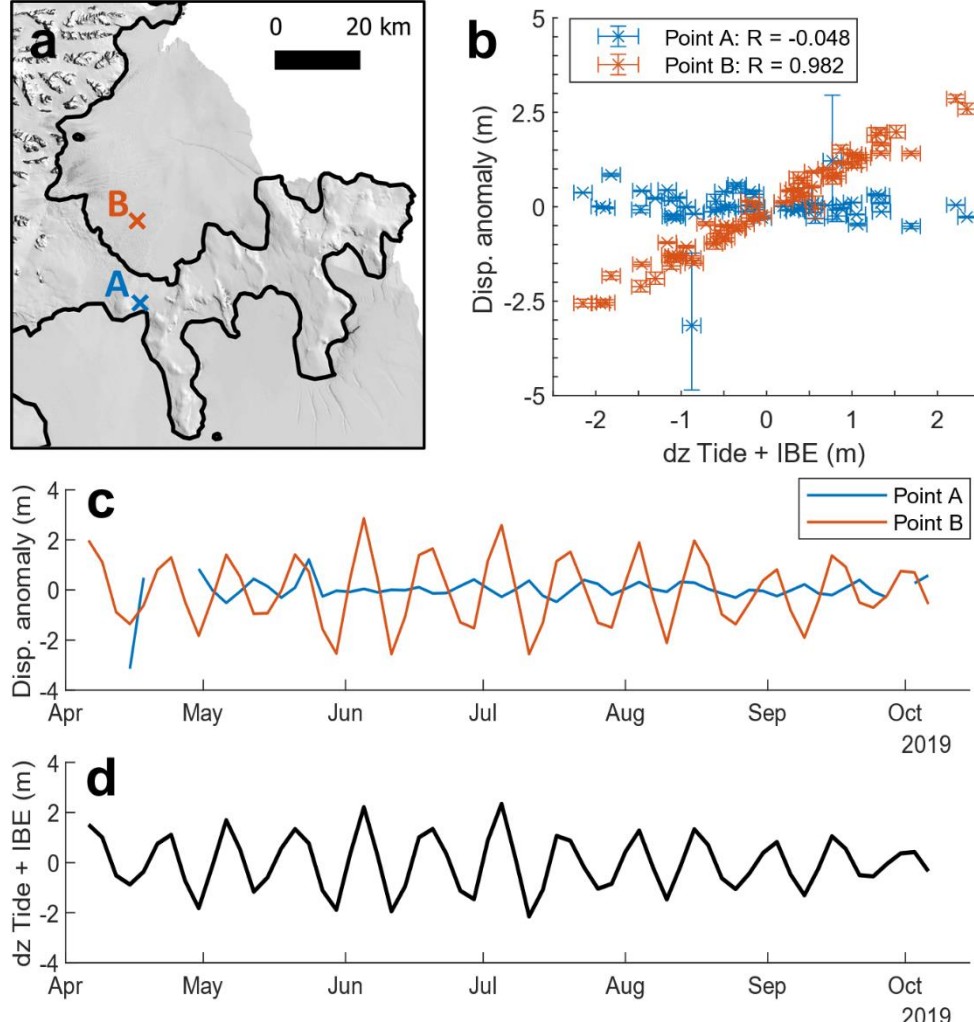

**Figure 1:** Measurement principle of the tidal motion offset correlation (TMOC) method. (a) Location of points A and B on the Jason Peninsula and the Larsen-B remnant respectively. The base-map is the LIMA Landsat mosaic (Bindschadler et al., 2008), with the MEaSUREs Antarctic boundaries v2 grounding line (black line) (Rignot et al., 2016) and sea-mask (Gerrish, 2020). (b) Differential tide plus inverse barometer effect plotted against the range direction anomaly for points A and B using ice velocity measurements from April 2019 to October 2019. (c) Time-series of range direction displacement anomaly for points A and B for April 2019 to October 2019. (d) Time-series of differential tide plus inverse barometer effect for April 2019 to October 2019 for point A.

series of modelled tide amplitude with a correction for atmospheric pressure; where these time-series have a strong and statistically significant correlation we conclude the ice must be floating and hence is seaward of the grounding line.

As in the DROT case, the apparent range direction displacement for a vertical displacement $\Delta\zeta$ and incidence angle $\theta_i$ is given by (Joughin et al., 2010b; Marsh et al., 2013; Friedl et al., 2020):

$$\Delta S = \frac{\Delta\zeta}{tan\theta_i} \tag{1}$$






In this equation, $\theta_i$ is dependent on the location of the satellite and surface topography. However for a specific location where SAR viewing geometries are approximately fixed (i.e. a time-series of a single Sentinel-1 frame), this is a linear function of vertical displacement. Therefore, we can calculate correlation between range direction ice speed and atmospherically-corrected tide height without the need to calculate the expected tidally induced velocity signal.

140       Figure 1 demonstrates the principles of the TMOC method for two points, Point A on the grounded ice of the Jason Peninsula and Point B on the Larsen B Ice Shelf remnant in the Scar Inlet (Fig. 1a). Comparing anomalies in range direction displacement in 2019 to modelled differential tide height with an inverse barometer effect (IBE) correction derived from ERA5 hourly sea-level pressure (Hersbach et al., 2023, 2020), we find that displacement anomalies at the floating Point B are strongly correlated ($R = 0.982$, $p < 0.05$) with differential tide amplitude between the SAR image acquisitions used in the offset tracking

result (Fig. 1b). This correspondence is also shown by comparing the time series of range displacement anomalies (Fig. 1c) to the time series of differential tide height (Fig. 1d).

## 2.2 Algorithm Description

The input data required for the TMOC GL method are time-series of ice speed observations in the range direction, modelled tide amplitude and sea-level atmospheric pressure. We measure range direction ice speed using frequency domain cross

correlation feature tracking between Sentinel-1 synthetic aperture radar (SAR) interferometric-wide (IW) mode images. This tracking is performed in range/azimuth radar geometry and we take the range component of these results and geocode the results at 100 m postings using the 200 m REMA DEM (Howat et al., 2019, 2022). For tidal motion, we use the CATS2008 Antarctic tide model, an update to the model described in Padman et al. (2002) (King et al., 2011), which includes tides in ice shelf cavities, to model tide displacement at 1 km resolution over the whole SAR image domain at the time of every SAR

acquisition used in the range speed timeseries. We interpolate these tide displacement predictions to the 100 m posting grid used for ice speed measurements using linear interpolation within the CATS2008 domain and nearest neighbour interpolation outside to produce a tide prediction for the entire image domain, including ice that is grounded in the CATS2008 model. For sea-level atmospheric pressure we use ERA5 hourly sea level pressure data (Hersbach et al., 2020, 2023), which we also interpolate to the 100 m ice speed grid for the time of each Sentinel-1 acquisition.

160       We form a time-series of range direction ice speed at each grid location from all available 6- and 12-day Sentinel-1 tracking pairs acquired between 1st April and 1st October of a given year or combination of years. Speed measurements are according to their temporal baseline to account for different tracking pair separations. We select SAR images from the winter months only to avoid the influence of the summer melt season, where surface melt between image acquisitions can cause vertical displacement of the radar scattering horizon within the firn pack leading to a measurement artefact that appears similar

to the range direction motion we seek to detect (Rott et al., 2020). We also form a time-series of the differential tide height, plus a correction for the inverse barometer effect at 1cm/kPa (Padman et al., 2003). We calculate the Pearson correlation coefficient between these two time-series at every grid point in the Sentinel-1 frame, along with the significance value of this





correlation. This procedure produces maps of correlation coefficient ($R$) between range direction ice speed and tide plus IBE vertical displacement, and the significance value of this coefficient $P_{corr}$.

We post-process both maps using a Butterworth lowpass filter of order two followed by an adaptive Wiener filter to remove high frequency noise from the results and remove pixels with fewer than 10 velocity tracking results. To account for the statistical significance of the correlation result we multiply the correlation coefficient map by $(1 - P_{corr})$ to calculate a significance adjusted $R'$; where the statistical significance of the correlation is high this does not substantially modify $R$ but does filter out high correlation values with low significance. We then form a mosaic of the scaled correlation coefficient map

for the Antarctic Peninsula region from 24 Sentinel-1 frames and produce a grounding line position by contouring this mosaic at a threshold of $R'>0.1$ while masking areas above 200 m altitude in the REMA Antarctic DEM. This threshold is chosen as a compromise between sensitivity and measurement noise. In the absence of any measurement noise a threshold of 0 should correctly map the GL at the most inland position, however, choosing 0 retains noise in the GL delineation, particularly in low tide amplitude areas such as the AP's west coast, so instead we use 0.1 to give smooth delineation of the GL. After contouring,

we merge adjacent contour lines and remove isolated inland points to produce the final GL dataset. In the idealised case with zero measurement noise and no phase shift between tidal amplitude and ice motion, the zero contour would give the inland limit of flexure, point F, however due to measurement noise and contouring at a value of 0.1 we expect that the chosen GL location will be slightly seaward of point F, but substantially closer than the ~ 1km seaward bias expected from DROT (Friedl et al., 2020) due to the use of a time-series approach and a far greater number of observations.

**3 Antarctic Peninsula Grounding Line for 2019-20**

**3.1 TMOC grounding line mapping**

We use the TMOC algorithm to process all Sentinel-1 frames acquired between April 1st 2019 and September 1st 2019, and April 1st 2020 and September 1st 2020, over the Antarctic Peninsula. This covers the region from Joinville Island at the northern tip to the southern edge of George VI Ice Shelf on the west coast, and to the edge of the Ronne Ice Shelf on the east coast (a

full list of Sentinel-1 frames used is given in Table S1). We used this data to produce a map of tidal motion correlation for the Antarctic Peninsula, with a time stamp of 2019 – 2020 (Fig. 2). For all measurements with respect to the coastline, we use the British Antarctic Survey Antarctic Coastline 7.2 vectors, released May 2020, for a contemporary coastline to our measurements (Gerrish, Laura, 2020).

We use this dataset to automatically delineate 9,300 km of the AP's grounding line and resolve all the AP's ice

shelves, including the small shelves and shelf remnants such as Seal Nunataks (Fig. 2b), Larsen-B remnant (Fig. 2c.), Muller Ice Shelf on the Arrowsmith Peninsula and minor ice shelves on Alexander Island's Beethoven Peninsula. In general, correlation values are highest on the eastern ice shelves, with values above 0.9 for Larsen-B remnant, Larsen-C and Larsen-D Ice Shelves, while values are lower on the western ice shelves, between 0.6 and 0.8 for Wilkins, Bach, and George VI Ice





**Figure 2.** (a) Map of TMOC significance adjusted tide correlation ($R'$) for the Antarctic Peninsula study region. Zoomed in maps show detail in (b) The Seal Nunataks and Larsen-B Inlet, (c) SCAR inlet, Larsen-B Ice Shelf remnant and Jason Peninsula, (d) Larsen-D Ice Shelf, and (e) George VI Ice Shelf and Alexander Island. All with the BAS 2020 coastline sea-mask (Gerrish, 2020).



Shelves. In addition to the GL, on ice shelves on both sides of the AP we are able to resolve the boundaries and therefore areas

of islands, nunataks, ice rises and pinning points with a high level of detail. For example, we delineate all of the Seal Nunataks

(Fig. 2b) except the smallest, Åkerlund Nunatak, which is only 100 m wide – the size of one pixel at our mapping resolution.

On Larsen-C, we map large islands such as Francis and Tonkin Islands, ice rises including the important Bawden and Gipps

Ice Rises, which mark the eastward extent of the shelf, and small nunataks like the 0.3 km$^2$ Table Nunatak, adjacent to the

Kenyon Peninsula. On the west coast, despite lower tidal amplitudes we also successfully resolve small features on George VI

Ice Shelf including the Martin Ice Rise and the southern margin pinning points around Eklund Islands (Fig. 2e.). On Wilkins

Ice Shelf we resolve dense and complex clusters of pinning points, such as the Petrie Ice Rises.

Although most glaciers on the peninsula outside the ice shelves are thought to be tidewater glaciers, which calve at

their grounding line and have no floating tongue, we can resolve floating sections of a number of glaciers on both sides of the

peninsula and measure the length of their floating tongues. On the east coast, glaciers with floating tongues are observed in

the embayments of disintegrated ice shelves, such as in the Larsen-B embayment, where Crane Glacier (4.0 km tongue) and

the Hektoria-Green-Evans (HGE) glacier system (12.3 km) are located, and in the Larsen-A embayment the Edgeworth-

Bombardier-Dinsmoor (EBD) Glacier system (10.7 km). On the west coast of the AP, we detect floating termini at Hoek

Glacier (1.4 km), Splettstoesser Glacier (1.7 km), and the Fleming-Airy-Seller glacier system (8.8 km), and on Alexander

Island's north coast, Hampton Glacier (6.0 km) and Roberts Ice Piedmont (4.3 km). On this coast, the small ice shelf (6 x 2.5

km) of Cadman Glacier is not well resolved and only has patches of significant correlation close to the GL, however, we

attribute this to the rapid acceleration (1,000 m/yr) that the glacier and ice shelf underwent in 2019 (Wallis et al., 2023b).

The TMOC method is also suitable for the identification of ephemerally grounded features; i.e. those which are

grounded at low tidal amplitudes but which can become ungrounded at high tides. In floating areas where there is a local

minimum of tidal correlation, but not reaching our 0.1 threshold for grounding line delineation we interpret these as sites of

ephemeral grounding. Examples of such features are seen around the grounding zone of the Larsen-B remnant close to Leppard

and Flask Glaciers (Fig. 1c) and along the east and west margins of George VI Ice Shelf (Fig. 1e).

### 3.2 Evaluation and Intercomparison

To evaluate the performance of the TMOC method, we directly compared contemporaneously produced TMOC and DInSAR

grounding line products. We produced differential interferograms from Sentinel-1 acquisitions in 2019 (Table S2) covering

the Larsen-B remnant, Larsen-C, Larsen-D and George VI Ice Shelves, and we manually delineated the inland limit of the

grounding zone using established methods. A comparison between the TMOC and DInSAR GL datasets in a region of

relatively static ice on Larsen-C Ice Shelf in the Stratton Inlet shows the performance of the TMOC algorithm and the

characteristics of the output datasets at the local scale (Fig. 3). This comparison shows that the TMOC GL location agrees very

well with DInSAR for the location of the GL and also is able to resolve an ice rise in the north margin of the Stratton inlet

where the fringe pattern is ambiguous and, therefore, accurately identifying this feature from DInSAR alone would be difficult.

InSAR coherence (Fig. 3c) is also a useful quantity for identifying grounding zone features as it is often high on grounded ice





**Figure 3: Comparison between DInSAR interferograms and InSAR coherence and TMOC significance adjusted tide correlation (R').** (a) Map of highlighted position (red box) for (b), (c), (d), on the Antarctic Peninsula, with the LIMA Landsat mosaic used as a basemap (Bindschadler et al., 2008), and BAS 2020 coastline and sea-mask (Gerrish, 2020). (b) Sentinel-1 DInSAR interferogram from September 2019, where the manually delineated grounding line from this interferogram is shown (black line). (c) InSAR coherence for one component interferogram from (b), also with manually delineated grounding line from (b) (black line). (d) TMOC significance adjusted tide correlation (R') and automatically delineated TMOC grounding line (white line). (e) Profile (A-A') of TMOC significance adjusted tide correlation (R') (d) (red line) and InSAR coherence (c) (blue line) , with the location of the automatically delineated TMOC GL also shown (black dashed line).

and ice shelves in hydrostatic equilibrium, but much lower in the grounding zone where the ice is deforming and displacing due to tidal motion (Gray et al., 2002). We extracted a transect through the grounding zone of the Stratton Inlet from both the

InSAR coherence and TMOC tide correlation data (Fig. 3e), which illustrates that the grounding zone features identified by



DInSAR are also resolved by the TMOC method. The transect also illustrates that the TMOC GL method successfully identifies grounding zone point F (Fig. 3e) as the tide correlation begins rapidly increasing from its minimum at the same point that InSAR coherence begins to fall, although the requirement to contour our results at R = 0.1 for reliable performance means that our method places the GL slightly seaward of point F.

We evaluated the performance of our TMOC algorithm quantitatively through a comparison to grounding line locations derived from a number of DInSAR datasets in five sectors of the AP region: the Larsen-B remnant, Larsen-C Ice Shelf, Larsen-D Ice Shelf, and the east and west sides of George VI Ice Shelf. For contemporaneous **GL** measurements, we use manual GL delineations of differential interferograms generated for this study from Sentinel-1 acquisitions in austral winter 2019. This data is impacted by low coherence in some areas and coverage is limited to the eastern margin of the Larsen-B

remnant, most of Larsen-C excluding some of the faster glaciers, a section of Larsen-D and the western margin of George VI Ice Shelf. For approximately contemporaneous GL measurements produced independently of this study we use the European Space Agency (ESA) Antarctic Ice Sheet Climate Change Initiative (CCI) project GL location data, choosing the most recent data for each comparison area (ESA Antarctic Ice Sheet Climate Change Initiative, 2021). This provides GL measurements from 2015-2017 covering the Larsen-B remnant and Larsen-C, excluding some fast glaciers, however no data is available for

Larsen-D Ice Shelf or George VI Ice Shelf. We also compare our TMOC results with the MEaSUREs Antarctic Boundaries Version 2 (MABv2) GL (Rignot *et al.*, 2013, Mouginot *et al.*, 2017) which is a composite of DInSAR GL measurements from 1992 to 2015 complemented by other GL measurements to provide a continuous GL around the Antarctic Ice Sheet. The advantage of intercomparing our TMOC GL to the MABv2 GL is that it provides full coverage across the study region and is the Antarctic GL most commonly used in the glaciology community. However, in regions with persistently low SAR

coherence, such as fast flowing glaciers feeding ice shelves and the eastern margin of George VI Ice Shelf, these DInSAR measurements are notably out of date because they were produced using data acquired during the ice or tandem phases of ESA's ERS-1 and ERS-2 satellites from 1992 to 1996, when shorter 3 and 1-day repeat periods ensured higher coherence.

        To perform the intercomparison between our TMOC method and these three DInSAR GL datasets, we label the line with the most complete coverage as the 'reference line' and the other the 'test line'. We measure the shortest distance from the

test point to the reference line at 1 km spaced points, and we calculate the sign (seaward or inland) of this distance by comparing the directions of the vector between the test point and the closest location on the reference line. This procedure produces a distribution of signed offsets between the two datasets which are averaged, with the standard deviation also calculated (Fig. 4, Table 1).

        This inter-comparison shows that the highest agreement is found between the TMOC and MABv2 grounding line

along the western boundary of George VI Ice Shelf, where the mean seaward offset is 0.6 m. The lowest agreement is measured on the Larsen-C Ice Shelf where mean seaward offset between the TMOC and MABv2 GL's is 262 m. The distribution of offsets is also variable between the different comparison areas, with minimum standard deviation of 415 m for the Larsen-C GL and maximum of 1450 m for the eastern margin of George VI Ice Shelf. On George VI Ice Shelf this distribution of offsets is dominated by a small number of regions where there are large ambiguities around relatively localised features (Fig. 4e, 4f).



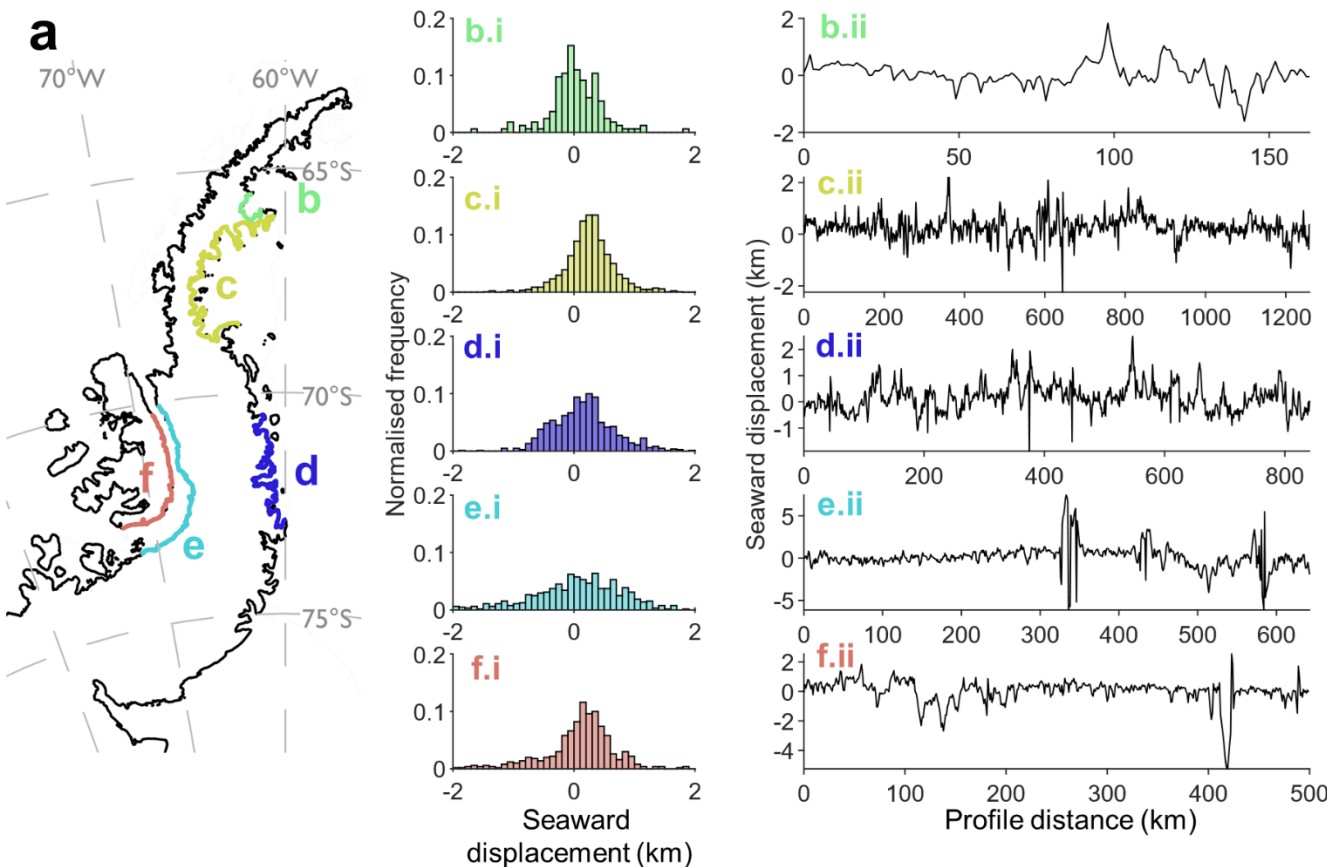

**Figure 4: Comparison between MEaSUREs Antarctic Boundaries v2 grounding line (MABv2) and TMOC 2019-20 grounding line for 5 locations:** (b) Larsen-B Ice Shelf remnant, (c) Larsen-C Ice Shelf, (d) Larsen-D Ice Shelf, (e) George VI Ice Shelf east margin, (f) George VI Ice Shelf west margin. (b.i – f.i) Histograms show the offset between the GL products produced by both methods, seaward displacement of the TMOC grounding line is sampled at 1km intervals. (b.ii – f.ii) Profiles of seaward displacement of TMOC grounding line at 1 km intervals along the sections coloured in (a).

This likely reflects the highly changeable surface conditions of this region, which causes low coherence in most interferograms and makes intensity feature tracking difficult, particularly on the ice streams of western Palmer Land. By contrast, on the Peninsula's eastern ice shelves there is a smaller standard deviation of the offsets between the TMOC and MABv2 GL. Here we find that the seaward offset of the TMOC GL with respect to MABv2 (mean 165 m) is positively correlated (R = 0.430 to 0.576, $p < 0.05$) with ice speed (Rignot, E. et al., 2017). This might be expected, as fast flowing glaciers tend to have the

thickest ice and deepest grounding zones, making tidally-induced vertical motion more complex, out of phase with tides, and difficult to detect against fast horizontal motion. Overall, in the whole intercomparison covering 3,409 km of AP coastline, the mean seaward offset and standard deviation of the TMOC GL compared to MABv2 is 164 m ± 803 m.

     Our comparison between the TMOC GL and contemporaneously acquired Sentinel-1 DInSAR GLs (CCI 2015-2017 and this study 2019), which have more limited spatial coverage, show a small seaward offset, where zero offset falls within

one standard deviation of the offset (Table 1). When the TMOC GL is compared to the 2015-2017 CCI GLs we measure





| Region | Tide GL vs MEaSUREs Antarctic Boundaries v2 GL | | Tide GL vs ESA CCI GL (2015-2017) | | Tide GL vs DInSAR GL (2019) | |
|---|---|---|---|---|---|---|
| | Mean seaward offset (m) | Std. (m) | Mean seaward offset (m) | Std. (m) | Mean seaward offset (m) | Std. (m) |
| Larsen B remnant | 36.0 | 434 | 233 | 428 | 178 | 174 |
| Larsen C | 262 | 415 | 438 | 502 | 198 | 316 |
| Larsen D | 162 | 516 | - | - | 135 | 548 |
| George VI East | 138 | 1449 | - | - | - | - |
| George VI West | 0.6 | 836 | - | - | 158 | 263 |
| **Total** | 165 | 803 | 412 | 498 | 185 | 295 |

**Table 1: Quantitative intercomparison between the TMOC GL location and other comparable GL datasets** including, the MEaSUREs Antarctic Boundaries v2 (Rignot et al.,2016), ESA CCI (2015-2017) (ESA Antarctic Ice Sheet Climate Change Initiative, 2021) and contemporaneous DInSAR GLs produced for this study (2019). For the CCI GLs and DInSAR GLs a percentage coverage with respect to the TMOC tide GL is also given, and the standard deviation is of the distribution of seaward offsets.

seaward offsets of 233 ± 428 m for the Larsen-B remnant and 438 ± 502 m for the Larsen-C Ice Shelf, with no DInSAR coverage in the other comparison regions. A comparison of the TMOC GL to 2019 DInSAR GLs produced for this study gives the only truly contemporaneous comparison between our new technique and established methods, providing partial coverage for all comparison areas except George VI East. We observe smaller seaward offsets between TMOC and the 2019 DINSAR dataset in comparison to the CCI GL data, ranging from 135 ± 548 m for the Larsen-D Ice Shelf to 198 ± 316 m for the Larsen-C Ice Shelf.

In summary, we find that the TMOC grounding lines have a small seaward offset compared to DInSAR GLs in the Antarctic Peninsula region, ranging from 0.6 ± 836 m to 438 ± 502 m. In all comparison areas for all datasets except the Larsen-B remnant in 2019 DInSAR, we find that zero seaward offset is within the standard deviation of offsets. Assuming that the 2019 DInSAR GL is the best dataset to accurately validate the performance of the TMOC GL method, we estimate that TMOC places the GL 185 ± 295 m seaward of the DInSAR GL (point F) location.



## 4 Change in Grounding Line Location

### 4.1 Grounding Line Retreat

Our 2019-2020 TMOC dataset can be used to study change in grounding line position (Point F) on the Antarctic Peninsula in
locations where low DInSAR coherence and unfavourable satellite altimeter track spacings mean there has been a paucity of
GL measurements in recent decades. A key region in this regard is the north-east coast of the Antarctic Peninsula where the
Prince Gustav, Larsen-A and Larsen-B Ice Shelves collapsed or partially collapsed in the 1990s and early 2000s (Rott et al.,
1996; Rack and Rott, 2004; Rignot et al., 2004; Cook and Vaughan, 2010). The glaciers in these former ice shelf embayments
have exhibited ice dynamic variability in the 20 years following the collapse, with satellite observations showing both ice
speed and surface elevation change (Rott et al., 2011, 2018). GL position measurements in this period of dynamic change have
been sparse for the major glaciers, including Crane and the Hektoria-Green-Evans (HGE) Glaciers in Larsen-B, Drygalski and
the Edgeworth-Bombardier-Dinsmoor Glacier system in Larsen-A and Sjörgen Glacier in the Sjogren Inlet. The most recent
GL measurements for any of these glaciers in either the MEaSUREs or CCI datasets is from 1996, and from static methods
there is the ASAID GL dataset which corresponds to Landsat-7 images from 1999-2003. A more recent study produced GL
locations for HGE in 2013 and 2016 and Crane Glacier in 2016 by interpreting surface slope and surface elevation change
from InSAR DEMs (Rott et al., 2020).

       We compared our 2019-20 TMOC GL to the composite MEaSUREs Antarctic Boundaries v2 (MABv2) GL dataset
(Mouginot et al., 2017) (Fig. 5, Table S3) and MEaSUREs timestamped DInSAR GL (Rignot et al., 2016) (Fig. S1, Table S3)
to measure grounding line change. The largest GL retreat occurred on the Hektoria-Green-Evans Glacier system (Fig. 5b),
where the GL of Hektoria Glacier has retreated along the central flowline by 16.3 km since 1996, and GL retreat of 11.7 km
is measured between the 2019-20 TMOC and the MABv2 GL data, which does not have a singular timestamp. Our TMOC
results also show that the GLs on both Green and Evans Glaciers' have retreated by 9.3 km and 6.7 km respectively relative
to their 1996 position. Our results on Hektoria Glacier are in agreement with 2016 GL measurements from a more recent study
(Rott et al. 2020), which show a maximum inland GL retreat of 0.5 km in comparison to our TMOC data. On Crane Glacier
the changes are more complex. The TMOC measurements show that the 2019-20 GL has retreated by 3.6 km compared to the
1996 DInSAR position, but is 1.0 km advanced from the MABv2 GL. Further north in the Larsen-A Embayment we observe
that the GLs of Edgeworth and Dinsmoor Glaciers (Fig. 5c) have retreated by 9.1 km and 4.2 km compared to the MEaSUREs
Antarctic Boundaries v2 GL (no timestamped DInSAR was available) (Table S3). Finally, outside of the north-east Peninsula,
we also observe GL retreat of Vivaldi Glacier in the Schubert Inlet of Wilkins Ice Shelf, where the GL has retreated by 2.2 km
compared to 1995 DInSAR. This result at Vivaldi Glacier is confirmed by comparison to automatically delineated 2019
DInSAR GLs (Mohajerani et al., 2021), which are available in this area and show the same pattern of change.



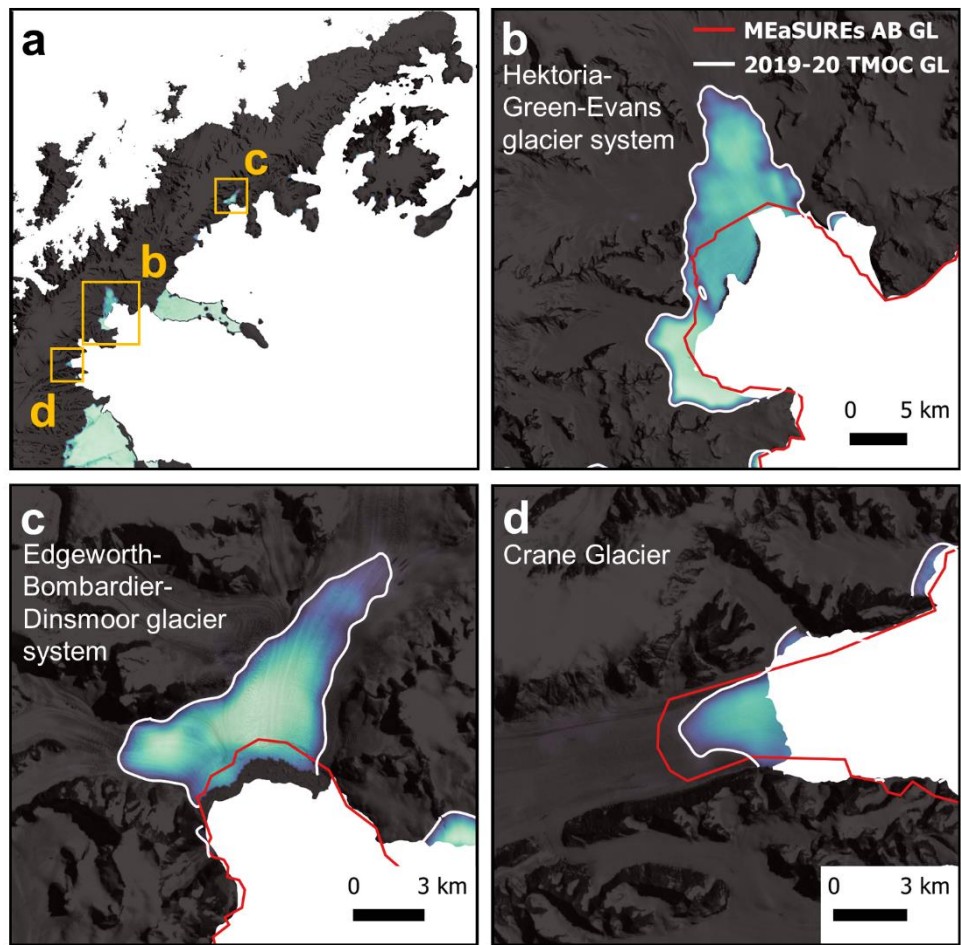

**Figure 5: Examples of grounding line change on the north-eastern coast of the Antarctic Peninsula.** (a) Summary map of the northern Antarctic Peninsula where the map shows the TMOC significance adjusted tide correlation (R'). (b-d) Zoomed in map of Hektoria-Green-Evans Glacier system (b), Edgeworth-Bombardier-Dinsmoor Glacier system (c), and Crane Glacier (d). In all maps the LIMA Landsat mosaic (Bindschadler et al., 2008) is used as a background image, with the BAS 2020 coastline (Gerrish, 2020), MEaSUREs Antarctic Boundaries v2 GL (red line) and TMOC 2019-20 grounding line (white line) also annotated.

## 4.2 Pinning Points and Grounding Zone Features

Our TMOC dataset is also suitable for the study of pinning points and ice rises underneath ice shelves (Fig. 6). We evaluate
the performance of the TMOC method by comparing pinning points and ephemeral grounding features in our dataset to pinning points in the MEaSUREs Antarctic Boundaries v2 (MABv2) grounding line dataset (Mouginot et al., 2017). We classify pinning points in one of three ways based on their visibility in the two datasets, InSAR only, Tide only or InSAR and Tide. We apply a more stringent criteria to tide only points than InSAR and tide, requiring them to be delineated at the 0.1 correlation coefficient threshold we use to map the GL to be classified as tide only points, whereas for InSAR and tide points we classify



a local minimum in the tidal correlation located at an InSAR derived pinning point to be sufficient, (Fig. 6c). We do not include large islands nunataks, or very large ice rises in this analysis.

      Our results show that on the AP the TMOC data can be used to identify and map 112 pinning points, including 22 new features that are not included in the MABv2 GL dataset. Of the 114 pinning points in MABv2, 90 are found in our TMOC data and 24 are not, which may suggest that the missing features have either become ungrounded in the period between the

two measurements, or that the sensitivity of the TMOC method was not sufficient to detect them. An example of the good performance of the TMOC dataset for resolving pinning points can be found at the Mill Inlet on the Larsen-C Ice Shelf (Fig. 6c). At the southern side of the inlet our data resolves four pinning points found in the MABv2 GL, but also identifies two additional pinning points and shows the shape of these sub-ice shelf features. New pinning points are found on all of the Peninsula's major ice shelves, for example on Larsen-D in the Smith Inlet (Fig. 6d) we identify a series of pinning points

which were not identified in any MEaSUREs or ESA CCI DInSAR GL products. It is possible that these points were not previously identified using interferometric techniques because they are close to a region of high shear on the Larsen-D Ice Shelf which leads to low SAR coherence. Even in regions of low tidal amplitude on the Peninsula's east coast our method is still suitable for studying pinning points, for example at the southern margin of George VI Ice Shelf (Fig. 6d) where we identify new pinning points close to the grounding line of CryoSat Ice Stream. On George VI Ice Shelf we also found two pinning

points from the MABv2 dataset which were not detected by TMOC, however, following a comparison with our 2019 co-temporaneous DInSAR we were able to confirm that these features were not measured by this method either, suggesting its more likely that the ice shelf has become ungrounded in this location.

      In addition to pinning points appearing or disappearing, the change in area of pinning points and ice rises can also be measured by TMOC, which is a useful indicator of change in ice shelf thickness, where an increase in area corresponds to ice

shelf thickening and reducing area to ice shelf thinning (Miles and Bingham, 2023). A good example of this is shown by the TMOC data forSeal Nunataks (Fig. 6b), where the pinning point area has decreased in comparison to the MABv2 indicating the thinning of this ice shelf remnant. The reduction in pinning point area on the Seal Nunataks is also confirmed by comparison to the ESA CCI DInSAR GL data for 2017. While it is possible that some proportion of the area difference should be attributed to differences in measurement techniques between the two time periods, our peninsula-wide statistical analysis suggests that

the TMOC GL will have a small seaward bias in comparison to interferometrically produced datasets (Fig. 4), therefore, the inland retreat is more likely due to real glaciological change, which would also fit in with our broader knowledge of ice shelf collapse and thinning on the AP (Shepherd et al., 2003; Rack and Rott, 2004).





**Figure 6: Pinning points measured using TMOC and MEaSUREs Antarctic Boundaries v2 (MABv2) data.** (a) Map of TMOC significance adjusted tide correlation (R') and pinning points across the Antarctic Peninsula. Pinning points are classified according to whether they appear in: both MABv2 DInSAR and TMOC data (pink diamond), MABv2 DInSAR only (red triangle) and TMOC only (purple circle). (b-e) Maps of TMOC significance adjusted tide correlation (R'), TMOC 2019-20 grounding line (white line), MABv2 grounding line (red line), with a LIMA Landsat mosaic base-map (Bindschadler et al., 2008) and BAS 2020 coastline and sea-mask (Gerrish, 2020). Zoomed in locations featured: (b) Seal Nunataks, (c) Mill Inlet, Larsen-C Ice Shelf, (d) Smith Inlet, Larsen-D Ice Shelf, (e) Eklund Islands, George VI Ice Shelf.



## 5 Discussion

### 5.1 Comparison to other non-InSAR grounding line remote sensing methods

We have demonstrated the high spatial coverage and accuracy of our TMOC grounding line data through comparison to DInSAR GLs from a number of sources. DInSAR still provides the most accurate method of measuring the inland limit of grounding zone deformation (point F) due to its high sensitivity to vertical displacement and its fine spatial resolution, and therefore remains the benchmark against which any novel GL method should be evaluated. However, as there are many regions where DInSAR GL measurements have not been possible since the 1990s, it is essential to have other methods to map change in GL location that can be used where interferometric coherence is not present. On the Antarctic Peninsula, previous studies have used Pseudo Crossover Radar Altimetry (PCRA) (Dawson and Bamber, 2017, 2020), Repeat Track Laser Altimetry (RTLA) and crossover analysis (Li et al., 2020, 2022), and break in surface slope methods (Bindschadler et al., 2011; Hogg et al., 2018).

PCRA was used to measure the location of points F and H across the Antarctic Ice Sheet, including the Larsen-C Ice Shelf on the Antarctic Peninsula (Dawson and Bamber, 2017, 2020), using eight years of CryoSat-2 radar altimetry from 2010-2017. On Larsen-C this method had an average seaward bias of -0.1 km (i.e. a landward bias of 0.1 km) and a standard deviation in this bias of 1.2 km compared to ESA CCI DInSAR data, and coverage of the AP region was 11 %. Similarly, RTLA was used to measure the location of grounding zone features first on the Larsen-C Ice Shelf and Larsen-B Ice Shelf remnant (March 2019 – March 2020) and later across the AIS (March 2019 – September 2020) (Li et al. 2020). Compared to ESA CCI GL data in the Larsen-C region, these studies find a mean absolute difference in the location of point F of 0.39 km with a standard deviation of 0.32 km. A later comparison of the Antarctic wide ICESat-2 product to a dataset of automatically delineated DInSAR GLs found a mean absolute separation in point F of 0.02 km with a standard deviation of 0.02 km, both in the Larsen-C region and across the whole Antarctic continent. We find that the performance of TMOC GL versus those produced from DInSAR for the ESA CCI AIS project is comparable to the performance of PCRA and RTLA versus the CCI data, while providing improved coverage compared to these datasets. The TMOC mean seaward offset against 2015-2017 CCI GL data is 0.41 km ± 0.50 km, for PCRA this is -0.1 km ± 1.2 km and for RTLA 0.39 km ± 0.32 km.

The strengths of TMOC, DInSAR and these altimetry-based methods should be seen as complementary; PCRA and RTLA provide grounding line measurements at the most southerly margins of Antarctica's Ross and Filchner-Ronne Ice Shelves, where no Sentinel-1 SAR acquisitions can be made due to the satellite's orbital limit, and perform best in these locations where altimetry tracks are most densely concentrated. Altimeter methods do not perform as well at lower latitude locations, such as the Antarctic Peninsula, due to wider track spacings, which cause lower spatial resolution, but it is here where TMOC excels due to complete SAR coverage and the method's ability to resolve small grounding zone features in regions where low coherence makes DInSAR ineffective. In addition to the limitations imposed by the acquisition pattern of Sentinel-1, our TMOC method is less effective in regions with 12-day repeat periods because there are fewer measurements to form a time-series and the tidal range offset anomaly is halved in magnitude compared to the optimal 6-day repeat. With



the upcoming launch of Sentinel-1c the 6-day repeat acquisitions should be restored, but again this demonstrates the value of operational, short repeat period SAR data acquisitions over the ice sheets. The effectiveness of the TMOC method is also

impacted by the tidal range at the target location, where higher tide amplitudes deliver a better quality result. However, we still achieve good performance on the west AP where rms tide height is 40-50 cm compared to 90-100 cm on the east coast (Padman et al., 2002). Similarly, our method does not account for short term variability in ice speed, such as seasonal speed changes (Boxall et al., 2022; Wallis et al., 2023a), which create range velocity anomalies that confound the tidal signal. Close to the grounding zone of thick ice shelves and ice streams, the motion of the ice may be substantially out of phase with the

ocean tide away from the GZ and hence our nearest neighbour interpolation from the CATS2008 tide model may not be optimal. In future studies this method could be improved by incorporating an elastic or visco-elastic beam model to improve estimates of point F from the observed correlation and through improved filtering of short-term velocity signals based on knowledge of where seasonal velocity variation or major dynamic changes have occurred.

## 5.2 Benefits of TMOC grounding line data

Accurate and up-to-date GL locations are key not only as a measure of ice sheet and glacier stability, but also as parameters for the interpretation of other observations and as inputs to modelling studies. On the Antarctic Peninsula our new 2019-2020 TMOC dataset provides a complete grounding line for the region and updated data for many locations where GL position has not been measured since the late 1990s or early 2000s.

The need for up-to-date publicly-available GL datasets to aid the interpretation of other remote sensing measurements

is well documented in the scientific literature, for example, in the study of rapid dynamic change on Antarctic Peninsula glaciers (Tuckett et al. 2019). In this study, short-term increases in ice velocity were observed on five glaciers including Hektoria, Crane and Jorum Glaciers in the Larsen-B embayment. This change in ice speed was linked to the drainage of supraglacial lake meltwater to the glacier bed, providing a mechanism for AP glaciers response to atmospheric warming, similar to that which we know to be widespread on the Greenland Ice Sheet (Sundal et al., 2011). There was constructive

debate in the scientific literature (Rott et al. 2020) about whether the use of an outdated GL position may have affected the conclusions drawn (Tuckett et al. 2019). More recent GL positions measured using the surface slope of TanDEM-X DEMs and surface elevation change data between 2013 and 2016 showed that two thirds of ice velocity sample locations were located on floating rather than grounded ice, meaning that meltwater drainage from these locations could not lubricate the flow of ice by the proposed mechanism (Tuckett et al. 2019). An alternative GL measurement (Tucket et al., 2020), produced using the

surface slope of the Reference Elevation Model of Antarctica (average date 2015) (Howat et al., 2019), suggests conversely that the majority of the ice velocity sample locations were located on grounded ice. Furthermore, this follow-on study argued that velocity measurements on floating ice do not invalidate the proposed forcing mechanism. On Hektoria Glacier our TMOC 2019/20 GL measurements agree most closely with the TanDEM-X 2016 GL (Rott et al., 2020) (Fig. S1), although without a concurrent measurement, it is not possible to definitively know over which time period GL retreat to this position occurred.

Overall, it is clear the paucity of grounding line measurements contributed to the difficulty in analysing the dynamic behaviour



of glaciers on the AP in this case, and so there is a need for tidally-sensitive methods and more regular GL measurements in these rapidly evolving locations.

Grounding line position measurements are essential for calculations of ice sheet mass balance through the Input-Output Method (IOM) and for altimetry measurements because these calculations must differentiate between changes on the grounded ice sheet which contribute to sea-level rise and changes on the floating ice which do not. In the case of the IOM, the GL position is required to locate the fluxgate, where ice discharge across the GL is measured. If fluxgates are erroneously placed seaward of the GL, then discharge calculations will be inaccurate, as basal melting, surface mass processes on floating ice and incorrect ice thickness may impact the result. For altimetry studies of ice sheet surface elevation change or ice shelf thickness change, whether ice is floating, or not, greatly affects the data processing methods, such as the choice of which geophysical corrections should be applied to the point elevation data. Knowledge of the GL is also important for interpretation of observed surface elevation change, such as determining which areas are included in the drainage basin total mass loss, due to the fact that floating ice in hydrostatic equilibrium changes elevation by approximately a factor of ten compared to grounded ice. Updated GL measurements from this study will be useful inputs to future studies of this nature, which may improve the accuracy of AP ice mass balance estimates, contributing to a reduction in the uncertainty of sea-level rise contributions.

Finally, modelling studies also require the GL position as an input dataset for the initialisation of model domains when assimilating observational datasets. In the AP region there are a number of examples where modelling studies depend on accurate GL locations, including glaciological processes case studies (Surawy-Stepney et al., 2023), projections of future change under different warming scenarios (Barrand et al., 2013), and calculating glacier bed topography by modelling ice thickness (Huss and Farinotti, 2014).

## 5.3 Grounding line change as an indicator of ice dynamics on the Antarctic Peninsula

Updated 2019/20 grounding line positions for the Antarctic Peninsula from our TMOC method have shown that GL retreat has occurred since the last period of widespread GL mapping on the AP in the 1990s and early 2000s. The observed GL retreat was concentrated in the north-eastern sector of the AP in the embayments of the collapsed Prince Gustav, Larsen-A and Larsen-B Ice Shelves, where we observe a maximum GL retreat of 16.3 km since 1996 (0.68 km/yr) on Hektoria Glacier (Fig. 5b, Supplementary Fig. 1b). We also observe instances of GL retreat on the west coast of the Peninsula on Vivaldi Glacier feeding the Wilkins Ice Shelf and at Fleming Glacier, an observation which confirms the results of Friedl et al. (2018), and the loss of ice shelf pinning points at the southern margin of George VI Ice Shelf.

The GL change observed in the Larsen-A and -B Embayments is particularly noteworthy given the ongoing dynamic evolution of these glaciers in response to the collapse of Larsen-A Ice Shelf in 1995, Larsen-B in 2002, and more recent changes which have been observed through to, and after, the 2019/20 period (Ochwat et al., 2023; Surawy-Stepney et al., 2023). In the Larsen-B Inlet, Crane Glacier and the HGE system have differing evolutions since the collapse of the Larsen-B Ice Shelf. For HGE from 2011 to 2016, the grounded component of the glacier system thinned by up to 10 m/yr (Rott et al., 2018) and thinning continued from 2018 to 2021 (Needell and Holschuh, 2023; Ochwat et al., 2023). Crane Glacier also





thinned inland from 2011 to 2013, however, from 2013 to 2016 the lower portion of Crane Glacier thickened. This trend was
accompanied by a decrease in ice speed from 3.9 m/d in September 2011 to 2.4 m/d in October 2016 (Rott et al., 2018) and
this trend continued to 2021 (Needell and Holschuh, 2023; Ochwat et al., 2023). The differing grounding line position evolution
of the HGE system and Crane Glacier in this period offers an explanation for this divergent behaviour. Compared to the GL
measured by Rott et al. for 2016, we find the GL of the HGE Glacier system is approximately unchanged from 2016 to 2019/20,
but for Crane Glacier we find the GL has advanced by 4.5 km in this period. This measurement of advance of Crane Glacier's
GL should be interpreted with some caution, because it compares two different techniques measuring GL position change over
a relatively short period of time and both the DEM (Rott et al., 2020) and TMOC techniques may be less accurate than DInSAR
delineations from the 1990s. However, this grounding line advance is plausible because the bed of Crane Glacier is retrograde
in this section, becoming shallower in the seaward direction (Fretwell et al., 2013). We can interpret these different behaviours
as being possible evidence for two contrasting regimes: Crane Glacier, where the GL has been advancing on a retrograde bed
from at least 2016 to 2019/20, causing the lower glacier to thicken, and Hektoria-Green-Evans Glacier system, where the GL
has been approximately static or retreating over the same period.

Since the 2019/20 period, for which our TMOC GL is dated, the glaciers of the Larsen-A and -B Embayments have
continued to evolve, and our grounding line measurements provide insight into the causes of these changes. In January 2022,
the multi-year landfast sea-ice in Larsen B embayment disintegrated and this event was followed by a major retreat and
acceleration of Crane and the Hektoria-Green-Evans glacier system (Ochwat et al., 2023; Surawy-Stepney et al., 2023). The
largest retreat was observed on Hektoria Glacier, which retreated by approximately 25 km between March 2022 to April 2023.
Our GL position results show that the calving front position of Hektoria Glacier on 29th March 2023 was between 1 km and
2.5 km inland of the 19/20 GL position, demonstrating that the majority of this retreat occurred on floating glacier tongue. On
Crane Glacier the calving front position in March 2023 closely matches the 2019/20 GL, showing a similar total collapse of
the floating ice tongue. In the Larsen A inlet from 2020 to April 2022, the Edgeworth-Bombardier-Dinsmoor glacier system
also retreated by 8.3 km, with the three glaciers separating and the calving fronts of Dinsmoor and Bombardier retreating to
their 19/20 GL position. Notably in contrast, Drygalski Glacier in Larsen A embayment, where we do not detect any floating
ice, did not retreat significantly in this 2022/23 period, showing that large ice front retreat in this area was limited to glaciers
with substantial floating ice tongues. The similarity of the retreat of the Larsen B glaciers and the EBD system in the Larsen-
A inlet raises the possibility that, in addition to sea ice changes, larger scale ocean and atmospheric forcings also played a role
in the recent evolution of the Larsen A and B embayment glaciers.

Understanding how the grounding line position of glaciers in the Larsen-A and -B Embayments evolve during a large
ice dynamic event is important, because Crane Glacier's retreat after the collapse of Larsen-B Ice Shelf was posited as an
observational example of marine ice cliff instability (Edwards et al., 2019, Meredith et al., 2019, Oppenheimer et al., 2019,).
A study revisiting this event 20 years later (Needell and Holschuh, 2023) noted that there is disagreement regarding the position
of the pre-collapse GL between remote sensing DInSAR observations of tidal flexure from the period (Rack and Rott, 2004)
and post-collapse marine geophysical surveys (Rebesco et al., 2014) which affects the interpretation of Crane's 2002-2004




post-collapse retreat. The continued monitoring of these highly dynamic glacier systems, which are exposed to extreme forcing events, has the potential to provide insights into ice dynamic processes relevant to the future evolution of the whole Antarctic

Ice Sheet, but these observations will require accurate knowledge of the geometry of the system to be interpreted correctly and with confidence.

On the west coast of the AP, observations have shown that glaciers can be vulnerable to grounding line retreat across retrograde bed slopes or loss of grounding from bed ridges, causing ice flow acceleration, increased ice discharge and mass loss (Wallis et al., 2023b, Friedl et al., 2018). Studies have linked the retreat of glaciers on the west AP to forcing by warming

ocean waters, which enhance melt rates and can cause GL retreat through intensified melting at the grounding zone. On the whole AP, monitoring GL change can provide important insight to understand how ocean warming is affecting the evolution of glaciers and the remaining southerly ice shelves. Up-to-date GL position measurements can identify where glaciers are situated on retrograde bed slopes, such as Fleming Glacier, and hence are vulnerable to rapid grounding line retreat and ice mass loss, such as the 1 km/yr speed increase observed on Cadman Glacier in 2019 (Wallis et al., 2023b).

**5.4 Systematic monitoring of Antarctic grounding line change**

In this study we used 2 years of Sentinel-1 SAR offset tracking data covering the period 2019-20 to produce a grounding line dataset for the Antarctic Peninsula Ice Sheet. However the TMOC method could be extended to a time-series of annual GL measurements using individual years of Sentinel-1 data. Presently, this is not possible, because the TMOC algorithm requires a 6-day repeat period for optimum performance and the failure of Sentinel-1B in December 2021 increased

repeat times to 12-days across Antarctica. However, after the launch of Sentinel-1C, the TMOC method could be used to make systematic and continuous measurements of the AP GL, extending the data record and providing longer multi-year temporal baseline comparisons to monitor GL change in previously difficult-to-measure locations.

For full and comprehensive mapping of the grounding line of the Antarctic Ice Sheet and sub-regions, such as the Antarctic Peninsula Ice Sheet, the best results are likely to be achieved through the combinations of multiple datasets derived

from independent remote sensing measurements. Our TMOC method provides the capability to delineate GL location accurately and reliably through the direct measurement of tidal motion in regions of low, or no, InSAR coherence, addressing a major limitation of DInSAR methods. We recommend that in future, TMOC GL data can be used to complement DInSAR GL delineations by providing coverage and monitoring in regions of persistent low coherence. Combined with other non-SAR grounding line remote sensing techniques, such as RTLA and PCRA described above, this would provide the most accurate

dataset for monitoring grounding line and pinning point change across the AIS.

**6 Conclusions**

We have developed a tidal motion offset correlation (TMOC) method for measuring the GL position, using the correlation between anomalies from range direction offset featuring tracking in synthetic aperture radar imagery and modelled tidal



amplitudes. We apply this method to the Antarctic Peninsula where contemporary measurements of grounding line position
are sparse, and demonstrate that the method performs well compared to highly precise DInSAR GL measurements, with a
mean offset between these data of 185 m and a standard deviation of 295 m. Our results show that TMOC provides excellent
grounding line coverage in regions of persistently low SAR coherence and can detect small grounding zone features, such as
pinning points and nunataks. From this data, we produce a complete grounding line dataset for the Antarctic Peninsula from
the Ronne Ice Shelf to George VI Ice Shelf, including Alexander Island, valid for 2019/20.

530         Our results show that this grounding line dataset can be used to measure change in GL position on the Antarctic
Peninsula, and we report examples of grounding line retreat which are largely concentrated in the north-east sector of the AP,
including glaciers which formerly fed the Prince Gustav, Larsen-A and Larsen-B Ice Shelves, which remain dynamically
imbalanced. We find a maximum GL retreat of 16.3 km on Hektoria Glacier compared to the most recent DInSAR
measurements from 1996. Overall, our results demonstrate that TMOC is a powerful new method for measuring grounding
line location in an automated way, which addresses shortfalls in existing techniques in a complementary manner. The technique
has the potential to greatly enhance the availability of regular GL measurements, particularly in complex regions such as the
Antarctic Peninsula. When used alongside existing methods, such as DInSAR and repeat altimetry, it represents significant
progress toward the goal of persistent monitoring of change in the grounding line location on the Antarctic Ice Sheet.

*Code Availability.* The code to implement the TMOC method will be made available at a public repository upon acceptance
of this manuscript. For the purpose of review a version can be provided by the corresponding author upon request.

*Data Availability.* The Antarctic Peninsula tide correlation map and Peninsula-wide 2019-20 grounding line position data will
be made available at a public repository upon acceptance of this manuscript. For the purpose of review these can be provided
by the corresponding author upon request.

*Author Contributions.* B.J.W. and A.E.H. designed this study, B.J.W. developed the TMOC algorithm, wrote the code,
implemented the method and processed the ice velocity data to produce the TMOC data and grounding line. Y.Z. produced
the DInSAR interferograms and Y.Z. and B.J.W. produced the manually delineated DInSAR grounding lines. B.J.W. and
A.E.H. wrote the manuscript, B.J.W. produced the figures, and all authors contributed to scientific discussion and revisions of
the manuscript.

*Competing Interests.* The authors declare that they have no conflicts of interest.

*Acknowledgements.* The authors gratefully acknowledge the European Space Agency (ESA) and the European Commission
for the acquisition and availability of Copernicus Sentinel-1 data. Funding is provided to B.J.W. by the Panorama Natural
Environment Research Council (NERC) Doctoral Training Partnership (DTP), under grant NE/S007458/1. Funding is
provided to A. E. H., by ESA via the ESA Polar+ Ice Shelves project (ESA-IPL-POE-EF-cb-LE-2019-834) and the SO-ICE
project (ESA AO/1-10461/20/I-NB) which both are part of the ESA Polar Science Cluster; and from NERC via the DeCAdeS



project (NE/T012757/1) and the UK EO Climate Information Service (NE/X019071/1). COMET is the UK NERC Centre for the Observation and Modelling of Earthquakes, Volcanoes and Tectonics, a partnership between UK Universities and the British Geological Survey.




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



## Appendix A: Additional detail of Antarctic Peninsula grounding line product

To accompany this paper, we provide a new grounding line dataset for the Antarctic Peninsula produced from our TMOC data valid for the period 2019 to 2020. This data covers the mainland Peninsula and islands in contact with ice shelves. We use the TMOC algorithm as described using Sentinel-1 SAR data for the period 2019 to 2020. Where the TMOC method does not detect floating ice, for example at rocky coastline or on glaciers with no ice tongue, we use the British Antarctic Survey coastline data for 2020 (Gerrish, 2020) so that we may provide continuous coverage of the coastline. One exception is Cadman Glacier which accelerated dramatically in 2019, here we use a grounding line derived from the REMA DEM to provide the grounding line (Wallis et al. 2023b). Our grounding line data is provided as a continuous closed grounding line for analysis purposes and as a collection of the individual line segments with their sources included.