# Peer review of "Change in grounding line location on the Antarctic Peninsula measured using a tidal motion offset correlation method"

_EGUsphere, 2023_

## Author Comment (AC1)

**Response to reviewer comments for 'Change in grounding line location on the Antarctic Peninsula measured using a tidal motion offset correlation method' – EGUSPHERE-2023-2874**

Benjamin J. Wallis, on behalf of the authors.

We thank the editor and three reviewers for their time and effort in reading this manuscript and providing useful and insightful comments. We are grateful for the thoughtful and collegial nature of the discussion and feel that responding to the reviewers' comments has improved this manuscript in terms of scientific content and clarity.

The reviewers' comments and our responses have been combined in this single document. There are some cases where comments from reviewers and our responses overlap, so placing all the comments together improves clarity. The comments and responses are given in table format below. The line field refers to the comment's line in the original manuscript, while 'new line' indicates the position of the relevant changes in the updated manuscript. The proposed changes to the manuscript, figures, and supplementary material follow these tabulated comments. We have only included figures and figure captions which are subject to change in this document. Figures 2, 3, 4 and supplementary figure 2 remain unchanged.

Reviewer 1:

| ID | Comment | Line | Response | New line |
|---|---|---|---|---|
| 1.1 | The authors propose a new method for mapping grounding line, evaluate its precision, and apply to the Antarctic Peninsula to detect changes since the 1990s. They report O(10km) retreat in the Peninsula since 1996. The paper is well written, the results are significant, the analysis of error is good, I would however recommend reducing the discussion section a bit to remove many paragraphs of literature review, perhaps trim by 30%. Overall, I would recommend publication after minor revision. I am only curious why the Mohajerani et al. 2021 dataset from year 2018, Antarctic wide, was not used for comparison. | | We thank the reviewer for their time, comments, and constructive feedback. We have implemented the vast majority of their suggestions, as indicated below. | |
| 1.2 | Line 36: Not sure I agree with this definition of the grounding zone. The grounding zone defines the range of migration of the grounding line, whereas what is described here is the flexure zone, which is the zone over which ice adjusts to flotation. It would be good to agree on that nomenclature, and I would recommend that the authors adopts this one. | 36 | We have used the definition of the grounding zone defined by Fricker et al. 2009: *'The GZ is the region of the ice sheet straddling the GL, encompassing the transition from fully grounded ice to ice in hydrostatic equilibrium with the underlying ocean.'*

 We choose this definition because it encompasses all features associated with the transition from fully grounded to floating ice. | 37 |
| 1.3 | Line 52. One of the earlier demonstration of GL mapping with DInSAR was Rignot, J. Glaciol., 1996 and Rignot et al. Science, 1997; Rignot, Science, 1998. The earliest I know of that used differential interferometry (which Goldstein et al., 1993 did not) was: Hartl, P., K.-H. Thiel, X. Wu, C. Doake & J. Sievers 1994 Application of SAR interferometry with ERS-1 in the Antarctic — Earth Observ. Quarterly 43: 1-4, but this is not a peer review article .. Just FYI. | 52 | Thank you for these helpful clarifications.

 **Done.** We have updated the reference with Rignot et al. 1996 and Rignot 1998. | 57 |
| 1.4 | Line 60. I actually do not agree with this. See Rignot et al. GRL 2011. The inland limit of fringes, the limits where you see ice lift up from the bed is the grounding line. What you mention here at the hinge limit is a mathematical model fitting to the elastic deformation which relates to the point of hinging and is typically displaced 1 km upstream of the GL. I sort of abandoned this point F over time, because the only point that matters is G, and Rignot et al. GRL 2011 describes how it is deduced from Differential interferometry. | 60 | **Done.** We have changed this line to:

 *'The inland limit of these fringes denotes the grounding line (Rignot et al., 2011)'* | 64 |
| 1.5 | Line 61. SAR coherence can ALWAYS be maintained. The true limitation of the repeat pass of the satellite. | 61 | **Done.** We have amended the text to reflect that the requirement for coherence is within the constraint of missions' acquisition plans. Changed to: | 65 |

| | | | | |
|---|---|---|---|---|
| | | | *'While highly accurate, the major limitations of the differential SAR interferometry (DInSAR) technique for grounding line measurements are the requirements for coherence between SAR acquisitions and well resolved interferometric fringes. This must be achieved while operating within the constraints of current missions' acquisition plans.'* | |
| 1.6 | Line 68. DInSAR is not used to delineate F, but G. See Rignot et al. GRL 2011. Also see above comments. | 68 | **Done.** Changed to: '…which along with DInSAR GL fringe delineation are dynamic methods that locate grounding zone features by measuring vertical ice motion in response to short-term local sea-level variation.' | 73 |
| 1.7 | Line 71. Did Joughin really map "F"? | 71 | **Done.** We have changed the phrasing of this sentence to: *'A small number of studies have used a technique called differential range offset tracking (DROT), a dynamic technique which measures vertical tidal motion in SAR imagery through intensity feature tracking rather than interferometry, to map the GL on individual glaciers of interest in regions without interferometric coherence (Marsh et al., 2013; Hogg, 2015; Joughin et al., 2016).'* | 76 |
| 1.8 | Line 95. If you call it "new" method. I need to be sure. Unfortunately, Joughin et al. Geophys. Res. Lett. (2016) used feature tracking on Pine Island Glacier to delineate the grounding line. Please refer to that earlier article by Ian J. and explain what is truly new in your TMOC method, which I think is the time series concept. Correct? | 95 | The reviewer is correct; the time series component, the idea of measuring correlation with modelled tide are the new parts of this technique. We believe that the method is explained in sufficient detail. We have given clear explanations of how it has been developed based on earlier differential range offset tracking studies, such as the one by Joughin that the reviewer references. See sections 2.1, 2.2 and Figure 1. | 101 |
| 1.9 | Line 116. Yes. | 116 | No response required. | |
| 1.10 | Speckle tracking is 10 times less sensitive to vertical than phase. See documentation of that in Mouginot et al. Geophys. Res. Lett. 2019 (phase map of Antarctica). A factor 10 would speak more than "significantly less". | 125 | **Done.** Changed to: 'around an order of magnitude less sensitive' | 131 |
| 1.11 | Line 134. This applies only if Delta S is in ground range. In slant range, this formula is wrong. Please correct text accordingly to say "ground range". | 134 | **Done.** Thank you for pointing out this ambiguity in our phrasing. | 140 |
| 1.12 | Line 159. What reference pressure do you use? | 159 | We calculate the difference in ERA5 sea-level pressure between the Sentinel-1 acquisitions which make up the feature tracking pair. Therefore, we do not use a reference pressure. | 162 |
| 1.13 | Line 182. You expect .. is this evaluated later on? | 182 | This is evaluated in section 3.2. | 193 |

| 1.14 | Line 208. Who says they are only tidewater? | 208 | **Done.** This statement is based on existing grounding line datasets. We have added a reference to these:

*'Although, according to existing grounding line datasets (Bindschadler et al., 2011; Rignot et al., 2016; Mouginot et al., 2017), most glaciers on the peninsula outside the ice shelves are thought to be tidewater glaciers'* | 218 |
|------|---------------------------------------------|-----|------------------------------------------------------|-----|
| 1.15 | Line 219. Ref. needed. | 219 | **Done.** We have added a reference to Zhong et al. 2023, which describes remote sensing of ephemeral grounding points in detail. | 222 |
| 1.16 | Line 225. Please note that Mohajerani et al. Nature Sci. Rep. 2021 produced an Antarctic wide data set for year 2018 that includes the East coast of the Antarctica Peninsula. Why did not you use that dataset, also distributed at NSIDC, for comparison? Or could you add it? | 225 | The Mohajerani et al. 2021 dataset is an excellent and innovative product for grounding line delineations. We agree with the reviewer's suggestion that a comparison to this dataset would be a useful addition to this study.

We did not include this dataset in the original manuscript, due to the format that the data are provided from Mohajerani et al. 2021. For the quantitative performance evaluation of our TMOC method (Figure 4) we require a single grounding line position for comparison. However, the grounding line delineations from Mohajerani et al. 2021 are given as vectors of every GL delineated from individual 6 or 12 day Sentinel-1 pairs, so there are dozens of GL measurements and no definitive single grounding line for the period. To use this data in a fair comparison, as we have done with the other reference GL datasets, would require us to develop a method to evaluate the best GL from the Mohajerani set. Producing such a secondary product is out of scope for this study.

The Mohajerani et al. 2021 dataset was also not included in the qualitative comparisons between datasets (Supplementary Figure 2) because these comparisons focused on areas where the Mohajerani data does not have coverage. The northward limit of GLs in the Mohajerani dataset in the AP is the Larsen B ice shelf remnant in the SCAR Inlet. The lack of coverage north of this position is most likely due to the incoherence of Sentinel-1 6 day pairs in locations north of the SCAR Inlet.

However, as stated above, we do agree with the reviewer's suggestion that a comparison to this dataset would be beneficial. Therefore, we have added a new supplementary figure (Supplementary Figure 1) to the manuscript which makes a qualitative comparison between our data and the Mohajerani et al. 2021 data around the Jason Peninsula, George VI Ice Shelf and Larsen D Ice Shelf | 299 |

where both have good coverage. This comparison shows that our TMOC grounding line falls within the distribution of GLs for the year 2018 delineated by Mohajerani's method.

We have added the following text to section 3.2:

*'We make a further comparison between our 2019-20 TMOC GL and DInSAR using GL automatically delineated with deep learning from all available Sentinel-1 DInSAR from the year 2018 (Mohajerani et al., 2021). This analysis is limited to a qualitative basis, because the available data are the set of GL delineations from all available interferograms, so there is not a definitive GL for a quantitative comparison without a further manual interpretation. We find that our TMOC GL is almost always within the distribution of GLs from Mohajerani et al.'s data, between the most inland and most seaward measured locations (Supplementary Figure 1). This comparison also highlights where the TMOC method can produce a grounding line measurement where 6-day repeat Sentinel-1 DInSAR cannot, for example in on Flask and Leppard Glaciers in the SCAR inlet (Supplementary Figure 1b), and Western Palmer Land (Supplementary Figure 1d). These results further support our conclusion that the TMOC GL position is located slightly seaward of the DInSAR GL position, if the most inland limit of DInSAR fringes observed in a given period is considered to the best measurement of the grounding line position.'*

| 1.17 | Line 256. I am not sure this is true for the Mohajerani et al 2021 dataset. They include ALL of these areas. Can you please include? | 256 | There are substantial gaps in the Mohajerani et al. 2021 dataset in the eastern margin of George VI Ice Shelf, this is shown in the new supplementary figure 1 which we have included. | 266 |
|------|------|------|------|------|
| 1.18 | Line 310. Please add error bars on these retreat estimates. | 310 | **Done** | 335 |
| 1.19 | Line 405-444. This is more of a review that a new contribution. Is this necessary? You already covered this in the intro. The Same comment may apply to 5.3. I would recommend cutting a bit into this, remove lit. review and focus on new elements. | 405 | This section was included to highlight how a paucity of grounding line measurements in the Antarctic Peninsula has impacted the interpretation of observations in this region. This is extremely relevant because our measurements provide updated GLs which address this issue.

Furthermore, the continued evolution of glaciers in the Larsen A and B embayments has been the subject of several publications recently: (Ochwat et al., 2023; Sun et al., 2023; Surawy-Stepney et al., 2024) and we feel our results make a contribution to this exciting and lively debate. | 429 |

| | | | | For these reasons, we consider this discussion necessary and have chosen to retain this section. | |
|---|---|---|---|---|---|
| 1.20 | Line 540. This should be done PRIOR to acceptance. I am surprised that this journal does not impose this to be done while the paper is submitted. I have therefore, as a reviewer, no access at this information for evaluation. Disappointing. | 540 | | The tide correlation maps, grounding line data and an example of the code implementation were made available for the reviewer in the 'reviewer assets' section of the EGUsphere website. We apologize if the data availability statement was misleading in this regard. It was written before the upload of these reviewer assets.

**Done.** Data availability statement now reads:

*'Data Availability. The Antarctic Peninsula tide correlation map and Peninsula-wide 2019-20 grounding line position data will be made available at a public repository upon acceptance of this manuscript. For the purpose of review these can be provided by the corresponding author upon request are available under the EGUsphere reviewer assets.'* | 575 |
| 1.21 | Figures are good. Figure 5 should explain what year is used for the MEaSUREs grounding lines. | 324 | | Figure 5 uses the MEaSUREs Antarctic Boundaries version 2 grounding line (Mouginot et al., 2017) (MAB v2). We have added the correct reference to the caption to make this clear.

The MAB v2 grounding line does not have a definitive timestamp, because it is a composite of DInSAR GL measurements from 1992 to 2015 complemented by other GL measurements to provide a continuous GL around the Antarctic Ice Sheet. We discuss this distinction in detail in section 3.2, lines 250-260.

We choose to show the MAB v2 for comparison as it is the most popular grounding line dataset within the community. Supplementary Figure 2 shows the same regions as Figure 5 with a comparison to many timestamped GL measurements. | Fig 5 |

Reviewer 2:

| ID | Comment | Line | Response | New line |
|---|---|---|---|---|
| 2.1 | This paper presents a nice technique for providing much-needed measurements. | | We thank the reviewer for their positive comments. | |
| 2.2 | Technically this is one of the cleaner papers I have reviewed. There are a number grammar and style issues, which comprise most of my comments. While largely optional, the paper's readability would be improved by applying many of them. The authors may want to consider using something like Grammarly in the future, which would have fixed many of these minor issues. | | We are glad to hear that the reviewer was pleased with the technical content of the manuscript. We found their suggestions for readability to be helpful and have incorporated the majority of these, as indicated below. | |
| 2.3 | A minor point is I would like to have seen more detail on the chip sizes used to do the speckle tracking, which may affect the biases (or differences in measurement techniques – see comment about GL location below). | | **Done.** We have added this detail: *'This tracking is performed in range-azimuth radar geometry using a cross-correlation window size of 256 x 64 pixels (range x azimuth) and a step size of a quarter window.'* | 158 |
| 2.4 | The NISAR launch should happen in 2024. With the longer wavelength and finer resolution, there will be more glaciers where the phase can be resolved and unwrapped. In these cases, this technique could be applied using the phase in place of the offsets. A sentence or two making this point would be useful. | | **Done.** We have added the following to the discussion: *'The planned launch of new L-band SAR missions, NASA and ISRO's NISAR (Rosen et al., 2017; Das et al., 2022) and ESA's ROSE-L (Davidson et al., 2021), will also provide new opportunities for grounding line monitoring using DInSAR and the TMOC method presented in this study.'* | 545 |
| 2.5 | As I commented below, it's worth pointing out that any of these techniques only detect a signal that is the proxy for the grounding line. I don't know of any work that has shown that the point of actual ungrounding is definitively given by any specific threshold for DiNSAR or this technique. What is most important is that a consistent proxy is used so that apples-to-apples comparisons are made when inferring grounding line retreat. It would be good to make a point like this. | | In response to Reviewer 3, we have endeavored to give a more absolute assessment of the accuracy of the TMOC method. See response to comment 3.2. | |
| 2.6 | Line 20. Since you don't match the retreat numbers to specific glaciers, remove ", respectively". | 20 | **Done** | 20 |
| 2.7 | Line 25. In many cases the area above the grounding line is also dominated by longitudinal stresses and vertical shear has little effect. Lateral shear stress is often as if not more important than longitudinal stress in many cases on both sides of the grounding line. Maybe rephrase as the transition from the region influenced by basal shear stress to the region with no drag. | 25 | Thank you for this clarification. **Done.** Changed to: 'transition between an inland flow regime influenced by basal shear stress, and a frictionless floating ice flow regime' | 25 |

| 2.8 | Line 33: Here an numerous other places hyphenate "ice-sheet" when its used as an adjective "ice-sheet mass loss" but not when used as noun, "Antarctic Ice Sheet". | 33 | **Done** | 32 |
|---|---|---|---|---|
| 2.9 | Line 37 Add "," before "which" | 37 | **Done** | 37 |
| 2.10 | Line 41 "and the extent" break the sentence in two here. | 41 | **Done** | 41 |
| 2.11 | Line 43 "," before "which" | 43 | **Done** | 42 |
| 2.12 | Line 46 "this point is followed" | 46 | **Done** | 46 |
| 2.13 | Line 55 "ice-flow" | 55 | **Done** | 59 |
| 2.14 | Line 57 remote "state" the ice flow may be steady (the same over two periods), but not necessarily steady state (i.e, if the flow is causing thinning). | 57 | **Done** | 61 |
| 2.15 | Line 58. Break start a new sentence at ", with the remaining" and add "differential" before "vertical" | 58 | **Done** | |
| 2.16 | Line 62. The data can be coherent but not suitable for mapping the GL. For example, the data can be well correlated but if the fringe rate exceeds the spatial sampling, the fringes are aliased even though they may remain coherent.  Maybe say something about the data should be adequately sampled so that the fringes can be resolved. | 62 | **Done.** Changed to:

'requirements for coherence between SAR acquisitions and well resolved interferometric fringes.' | 64 |
| 2.17 | Line 71. A better reference than Joughin et al 2010b is Joughin et al 2016 https://doi.org/10.1002/2016GL070259 | 71 | **Done.**

Thank you for the helpful suggestion. | 77 |
| 2.18 | Line 78. Maybe say "slope-related shading effects" | 78 | **Done.** | 83 |
| 2.19 | Line 81. Strictly speaking Goldstein did not use DiNSAR since he only had a single interferogram. It happened to be one that with a relatively simple flow field so that GL was visible. | 81 | **Done.** Changed to:

'InSAR and DInSAR mapping of the GL in Antarctica has been carried out using SAR data since 1992 (Goldstein et al., 1993)' | 86 |
| 2.20 | Line 103: I think you mean "formerly" not "formally" | 103 | **Done** | 109 |
| 2.21 | Line 108: I think "sparse" is spelled the same way on both sides of the Atlantic (not sparce). | 108 | **Done** | 114 |
| 2.22 | Line 110: Add a "," after "methods" | 110 | **Done** | 116 |
| 2.23 | Line 117. In general its good to have at least an introductory sentence between Level 1 and Level 2 heading (i.e., before 2.1 Physical Basis). | 117 | In this case we have chosen to remain with our exiting formatting. | 125 |
| 2.24 | Line 121. Don't start a sentence with an acronym. | 121 | **Done.** Changed to:

'The DROT method' | 127 |
| 2.25 | Line 122 after "range direction" add "(line of sight)" | 122 | **Done** | 128 |
| 2.26 | Line 122 ", and the speed" start a new sentence here. | 122 | **Done** | 128 |
| 2.27 | Line 124 Don't' start with acronym. How about "There are several limitations with DROT;" | 124 | **Done** | 130 |

| 2.28 | Line 125 add "," before and after ", and hence, " | 125 | **Done** | 132 |
|------|---|---|---|---|
| 2.29 | Line 126. Be careful with terms like "feature tracking". You are actually using a combination of speckle tracking and feature tracking here. | 126 | **Done.** Changed to 'Intensity feature tracking' to be more specific. | 133 |
| 2.30 | Line 126 remove "and" | 126 | **Done** | 133 |
| 2.31 | Line 127 change "…motion," to "…motion; " | 127 | **Done** | 134 |
| 2.32 | Line 132 add a "," before "we conclude" | 132 | **Done** | 139 |
| 2.33 | Line 151 add "," before "and" or better yet make it a new sentence. | 151 | **Done** | 159 |
| 2.34 | Line 155 Consider breaking this long sentence in two. | 155 | Thank you for the suggestion, but we have chosen to leave this unchanged. | 164 |
| 2.35 | Line 161 Sentence "Speed measurements are …." Its not clear what is meant here. Rephrase. | 161 | **Done.** Changed to: ' *Speed measurements are scaled according to'* | 171 |
| 2.36 | Line 162-164. Remove "We select ….melt season" And start a new sentence first explaining melt effects. "Surface melt between…." Then a new sentence "Thus, we select only SAR images from winter periods" or something like that. | 162 | **Done** | 172 |
| 2.37 | Line 176 add a "," before "while" | 176 | **Done** | 186 |
| 2.38 | Line 177. I don't like the sentence "In the absence of noise…" Whenever you estimate a correlation, you get a random variable, not something that's precisely 0. Moreover, with no noise, you might pick up subtle speed variations that correlate with the tide. I am more surprised that the threshold is as low as 0.1. So you don't have to justify not using 0. Remove the sentences referencing 0. And just say something like "We found a threshold of 0.1 give a good balance between false and missed detections of grounded ice". | 177 | **Done.** We have removed this sentence as suggested. This now reads:

*'This threshold is chosen, because we find it gives a good compromise between sensitivity and measurement noise. After contouring, we merge adjacent contour lines and remove isolated inland points to produce the final GL dataset. In the idealised case with zero measurement noise and no phase shift between tidal amplitude and ice motion, the zero contour would give the inland limit of flexure, point F, however due to measurement noise and contouring at a value of 0.1 we expect that the chosen GL location will be slightly seaward of point F, but substantially closer than the ~ 1km seaward bias expected from DROT (Friedl et al., 2020) due to the use of a time-series approach and a far greater number of observations.'* | 190 |
| 2.39 | Line 186: Add introductory text before 3.1 | 186 | In this case we have chosen to remain with our exiting formatting. | 196 |
| 2.40 | Line 188: Read better if you say what "this" is. "This data set covers…" | 188 | **Done** | 199 |
| 2.41 | Line 190: "These data" not "this data" | 190 | **Done** | 201 |
| 2.42 | Line 203: Nothing wrong with "map" but how about "detect" | 203 | **Done.** Changed to 'resolve' | 213 |
| 2.43 | Line 209: Do you have a reference to backup this belief (i.e., most termini are grounded)? | 209 | **Done.** Added a citation to existing GL datasets:

(Bindschadler et al., 2011; Rignot et al., 2016; Mouginot et al., 2017) | 218 |
| 2.44 | Line 244: Add a "," before "we" | 224 | **Done** | 235 |
| 2.45 | Line 251 "," before "which" | 251 | **Done** | 262 |
| 2.46 | Line 263 Change "which" to "that" | 263 | **Done** | 273 |

| 2.7 | Line 270. How about "These ambiguous regions.." rather than just "This…" | 270 | **Done.** Changed to:

'These ambiguous regions likely reflect areas of highly changeable surface conditions, which cause…' | 281 |
|---|---|---|---|---|
| 2.48 | Line 274: Not sure why there is an initial in the Rignot citation. | 274 | **Done.** | 285 |
| 2.49 | Line 274: Change "This might be expected, as fast…" to "This might be caused because fast…" The issue is cause and effect concerning the glaciers, not what people might expect. | 274 | **Done.** Changed to: 'This may be explained because…' | 286 |
| 2.50 | Paragraph with Line 290. It might be good to say with either method a proxy for the grounding line is being used. No one can really say exactly where the grounding line is based on the fringes. Instead, what we have are educated guesses used to define conventions. The bias is not with respect to the true grounding line, but instead to the inland limit of fringes as used by MEASURES. | 290 | We have amended this section in response to Reviewer 3's comment that accuracy should be assessed in a more absolute way. See response to comment 3.2. | |
| 2.51 | Line 300: Spell out Grounding line when staring a sentence. | 300 | **Done** | 324 |
| 2.52 | Line 312 "Glaciers'" remove the "'". It is not needed as written. | 312 | **Done** | 336 |
| 2.53 | Line 312 "respectively" should be ", respectively," | 312 | **Done** | 337 |
| 2.54 | Line 316 "," before "we" | 316 | **Done** | 341 |
| 2.55 | Line 340 "which" to "that" | 340 | **Done** | 365 |
| 2.56 | Line 345 Break sentence "… by TMOC. Following a comparison with …. DInSAR, we were…" | 345 | **Done** | 370 |
| 2.57 | Line 360: Add a sentence or two before 5.1 | 360 | In this case we have chosen to remain with our exiting formatting. | 385 |
| 2.58 | ▪ Line 370: With short enough baselines so the phase could be unwrapped, you could apply your technique, which should be an improvement over single DInSAR. | 370 | This is an interesting idea, but we have chosen not to include it, as other reviewers already recommended shortening the discussion section. | 394 |
| 2.59 | ▪ Line 371: Rewrite to move acronym from starting the sentence. | 371 | **Done** | 395 |
| 2.60 | Line 530. Another overly long sentence break into 2 or even 3 pieces. | 530 | **Done** | 562 |

Reviewer 3:

| ID | Comment | Line | Response | |
|---|---|---|---|---|
| 3.1 | The proposed research approach is sound, interesting, and well-organized. | | We thank the reviewer for their positive assessment. | |
| 3.2 | However, I am afraid the authors may be, in some ways, overstating the results and the capabilities of the proposed method. For instance when they claim: "the method performs well compared to highly precise DInSAR GL measurements, with a mean offset between these data of 185 m and a standard deviation of 295 m" they asses the presence of a bias in their data, but they do not address the actual accuracy of the technique itself.

For instance, Single DInSAR Inteferogram grounding line measurements have an accuracy of about +/- 300 m when locating grounding lines.

The proposed technique could most likely map grounding lines within +/- 1-2 km accuracy. This is because of the Sentinel-1 pixel size, the accuracy of pixel tracking technique itself, and the inaccuracies due to the non-accurate CATS model near the grounding line together with the coarse resolution of IBE corrections. Finally, this technique assumes no horizontal velocity changes during the observation period which can also affect the correlation with tidal levels.

I would like these numbers to be specified in the text in order to better characterize the applicability of the presented technique. | | We acknowledge the reviewer's concerns that the accuracy of the method presented in this study should be described as comprehensively as possible. We have endeavored to do this transparently with the extensive intercomparison described in section 3.2.

We believe that the CATS2008 tide model and sea-level pressure data from ERA5 reanalysis represent the best publicly available input data for our method.

The limitation of not accounting for short term ice speed changes was already raised in the manuscript in Section 5.1 line 397. We have expanded this sentence to read:

*'…our method does not account for short term variability in ice speed, such as seasonal speed changes (Boxall et al., 2022; Wallis et al., 2023a) or rapid ice flow accelerations, which create range velocity anomalies that confound the tidal signal.'*

As the reviewer suggests we have included the measurement error associated with DInSAR grounding line delineations, using the figure of ± 100 m quoted by Rignot et al., 2011. Combining this uncertainty in quadrature with the bias plus standard deviation of our method gives an accuracy for our TMOC method of ± 490 m.

We have added the following text to describe this:

*'Assuming that the 2019 DInSAR GL is the best dataset to accurately validate the performance of the TMOC GL method, we estimate that TMOC places the GL 185 ± 295 m seaward of the DInSAR GL location. When the upper limit of this bias and variability is combined with a standard error of DInSAR GL delineations of ± 100 m (Rignot et al., 2011), we estimate the TMOC method can locate the grounding line position with an accuracy of ± 490 m.'* | 421, 311 |
| 3.3 | Most importantly, all the grounding line retreat rates of the same magnitude of the grounding line mapping accuracy (i.e. anything slower than 1 to 2 km/yr retreat) would lose statistical | | The reviewer makes a valid point that over a single year retreat rates less than the measurement uncertainty can not be resolved. However, it is the absolute grounding line position change between | 536 |

| | significance. This should be specified in the discussion and the conclusions. | | measurements, not the rate of change, which is the limiting factor. Retreat rates on the order of the measurement uncertainty per year could be resolved if they are sustained over a multi-year temporal baseline.

We agree that this point could be better developed and quantified in the manuscript, so we have added the following the discussion:

*'Mappings of GL position using the TMOC method would be suitable to measure changes in GL position which exceed the combined uncertainty of two TMOC GL measurements, excluding the seaward offset with respect to DInSAR GLs, which would be approximately constant between TMOC measurements. This gives TMOC the capability to resolve GL retreat rates of greater than 418 m/yr between measurements for two adjacent years or 83 m/yr if this were sustained for 5 years.'* | |
|---|---|---|---|---|
| 3.4 | Line 43-45 No reference to a figure where point F or G are located. | 43 | We made an active choice not to include a diagram very similar to those induced in many other papers on grounding lines (Smith, 1991; Vaughan, 1994; Fricker et al., 2009; Brunt et al., 2010, 2011; Dawson and Bamber, 2017). The readership of the Cryosphere who will be interested in this paper have already seen this diagram many times and our description does not add any new perspective from the diagrams in the referenced papers. Therefore, we do not believe it to be necessary or that it would benefit the manuscript to reproduce these figures verbatim.

We have instead added a sentence to explicitly refer the reader to the referenced material for a visual depiction and the majority of these article are open access:

*'Schematics showing the cross section of a grounding zone are widely available in the literature (Smith, 1991; Vaughan, 1994; Fricker et al., 2009; Brunt et al., 2010, 2011; Dawson and Bamber, 2017; Friedl et al., 2020)'* | 48 |
| 3.5 | Lines 72-74 The sensitivity of the pixel offset mapping method also depends on the Radar pixels size. Achievement displacement accuracies are usually 1/8 of the radar pixel size (De Zan 2014)
De Zan, F. (2013). Accuracy of incoherent speckle tracking for circular Gaussian signals. IEEE | 72 | **Done.** We have amended this line to read:

*'The performance of the method is dependent on the sensitivity of the offset tracking results, determined by the range direction pixel size; and the magnitude of the tide amplitude in the study region.'* | 77 |

| | | | | |
|---|---|---|---|---|
| | Geoscience and Remote Sensing Letters, 11(1), 264-267. | | | |
| 3.6 | Figure 1C is confusing, A straight line does not give the idea of when images have been acquired. | 131 | **Done.** We have added markers to this figure for clarity. | Fig 1 |

[revised manuscript text omitted]

**Amended supplementary material:**

[Figure]

**Figure S1.** Comparison between 2019-2020 TMOC grounding line (GL) and the set of grounding lines from Mohajerani et al. 2021. **(a)** Overview map of the Antarctic Peninsula. Coastline and grounding line from MEaSUREs Antarctic Boundaries v2 (Mouginot et al., 2017). **(b)** Insert of the Jason Peninsula, **(c)** Larsen-D Ice Shelf, **(d)** George VI Ice Shelf all showing the 2019-2020 TMOC GL (red) and the set of GLs from Mohajerani et al. 2021 (blue), Background image is LIMA Landsat mosaic (Bindschadler et al. 2008).

---

## Author Response (AR2)

**Response to second round of reviewer comments for 'Change in grounding line location on the Antarctic Peninsula measured using a tidal motion offset correlation method' – EGUSPHERE-2023-2874**

Benjamin J. Wallis, on behalf of the authors.

We thank the editor and reviewer for their time and effort in reading this revised manuscript. We are grateful for the reviewer's comments and have responded to them here.

The line field refers to the comment's line in the original manuscript, while 'new line' indicates the position of the relevant changes in the updated manuscript with tracked changes. There are no changes to the figures in this second set of revisions.

Reviewer 1:

| ID | Comment | Line | Response | New line |
|----|---------|------|----------|----------|
| 1.1 | The definition of the GZ adopts something written by H. Fricker, which refers to the ocean grounding zone or the flexure zone. It is a point of importance because some authors have mapped the extent of this zone and call it GZ width. Yet, the grounding line itself is migrating with the tide, atmospheric pressure and long term. How do you resolve that? How will you separate the fact that the flexure zone has some width, always is, typically 5 to 10 km, whereas the true grounding zone, which is the locus of temporal variations of the position of the GL, is something more fundamental, more new - because most folks did not have the data to look at it (See Mohajerani et al. 2021). It is fine to leave it like this, but as satellites provide more and more info on the position of the GL, i.e. the GZ, which is of high importance for model, their definition will become out of date and possibly misleading. Your choice. I am not sure my comment was understood. I hope this clarifies. I can see that lots of people will be - and have been - confused. | 36 - 54 | We thank the reviewer for clarifying their previous position and agree that grounding line/zone terminology is somewhat mixed within the community since the recent increase in interest in tidal grounding line migration.

To address this, we have clarified our definition to distinguish the 'flexure zone' where ice adjusts to HE excluding tides; the 'grounding line tidal migration zone' which is the locus of true GL locations due to tide and IBE; and the 'grounding zone' to mean the combination of these two.

We chose to maintain a broader definition of grounding zone, because it is useful to discuss 'grounding zone features' such as pinning points and is less in conflict with older definitions of the grounding zone. By specifying the flexure zone and tidal migration zone, we resolve any ambiguities.

We have also tightened up our use of the terms grounding zone and grounding line throughout the paper.

The paragraph introducing this terminology now reads:

*'Rather than having a fixed location, the grounding line is a transitory feature which constantly changes over short (daily) and longer term (decadal) timescales. It is located within a wider flexure zone (sometimes also called the grounding zone), which characterises the larger area (1 − 10 km wide) where the transition from grounded ice to complete hydrostatic equilibrium occurs (Brunt et al., 2010, 2011; Fricker et al., 2009; Smith, 1991; Vaughan, 1994). The flexure zone is made up of several features; the most inland of these is the landward limit of ocean induced ice flexure, point F, which is located slightly inland of the true GL, point G, due to the elastic properties of ice (Padman et al., 2018; Rignot et al., 2011; Vaughan, 1994). In the seaward direction this point is followed by the break in surface slope, point Ib, and the landward limit of stable hydrostatic equilibrium, point H. Additionally, in locations where there is an ice plain at the flexure zone, point Ib may be located inland of the GL, point G (Brunt et al., 2011; Corr et al., 2001). Schematics showing the cross section of the grounding line are* | 36 - 63 |

*widely available in the literature (Brunt et al., 2010, 2011; Dawson and Bamber, 2017; Fricker et al., 2009; Friedl et al., 2020; Smith, 1991; Vaughan, 1994). The true grounding line is a sub-glacial feature, so cannot be directly detected by satellite remote sensing measurements, which must instead measure surface expressions which are proxies for the GL or are used to deduce the GL position. Additionally, the true GL where grounded ice loses contact with the bed can migrate with changing sea-level caused by ocean tides and atmosphere pressure variations by the inverse barometer effect (IBE). This range of short-term tidal grounding line migration has also been referred to as the grounding zone by recent publications (Mohajerani et al., 2021; Rignot et al., 2024). The extent of this migration is also controlled by bed topography, ice thickness and ice rheology (Brunt et al., 2010; Jonathan and R, 1994; Padman et al., 2018) and further complicated by non-linear tidal migrations, which can show threshold and hysteresis behaviour (Freer et al., 2023; Milillo et al., 2022). For the purposes of this study we use the following terminology: 'flexure zone' to describe the features of ice flexure relation to the transition from grounded to hydrostatic equilibrium, excluding tides; 'grounding line tidal migration zone' (TMZ) to describe the locus of true grounding line migration due to tides and IBE; and 'grounding zone' (GZ) to encompass the combination of these. We use 'grounding line' (GL) to mean the inland limit of the grounding zone identified by remote sensing methods, as this is the focus of this study, and we are explicit about which grounding zone feature this refers to where required.'*

| 1.2 | Speckle tracking is 10 x times less accurate than phase mapping. This difference in performance has been thoroughly and extensively documented for velocity mapping in peer reviewed publications and is valid for a range of SARs. There is no reason to expect a difference in performance when mapping grounding lines, i.e. a differential motion. Speckle tracking has an intrinsic resolution of about 350 m because you have to average many pixels to get the offsets. The authors claim that they pick the GL within 400-500 m. This seems hard to believe and quite optimistic. I do not expect a precision to be better than 1 km, which is still useful | 311 | A very similar point to this was raised in the first round of review by reviewer 3. Please see the round of response to reviewers document comment 3.2. In response to this comment, we extended the discussion of errors in section 3.2 of the paper (reproduced from previous response to reviewers). We believe the changes made to reviewer 3's comments also adequately address this reviewer's comment.

Firstly, the reviewer's assertion that GL position could only be determined to within 1 km is most likely based on a single offset tracking result, ie in differential range offset tracking (DROT). We have explicitly acknowledged that offset tracking is less sensitive than DInSAR in the manuscript: *'There are several limitations of DROT; it is around an order of magnitude less sensitive vertical motion than* | |
|---|---|---|---|---|

*DInSAR'* (Line 130). Our approach, however, is based on a time-series of offset tracking results over two years, improving the quality of the measurement compared to DROT.

Secondly, the figure of 490 m uncertainty on the grounding line position is based on an extensive evaluation and intercomparison exercise described in section 3.2 of the manuscript. We believe this gives a transparent and fair evaluation of our method's performance against established datasets. Providing an uncertainty estimate based on comparison to contemporary DInSAR measurements gives readers an understanding of the performance of our method based in the accuracy of comparable established techniques. In our opinion this is the best way to communicate the performance of our method.

The original response to reviewer 3's first round comment is reproduced below:

*As the reviewer suggests we have included the measurement error associated with DInSAR grounding line delineations, using the figure of ± 100 m quoted by Rignot et al., 2011. Combining this uncertainty in quadrature with the bias plus standard deviation of our method gives an accuracy for our TMOC method of ± 490 m.*

*We have added the following text to describe this:*

*'Assuming that the 2019 DInSAR GL is the best dataset to accurately validate the performance of the TMOC GL method, we estimate that TMOC places the GL 185 ± 295 m seaward of the DInSAR GL location. When the upper limit of this bias and variability is combined with a standard error of DInSAR GL delineations of ± 100 m (Rignot et al., 2011), we estimate the TMOC method can locate the grounding line position with an accuracy of ± 490 m.'*

For these reasons we have not modified the manuscript in response to this comment.